



# Ice Anatomy: A Benchmark Dataset and Methodology for Automatic Ice Boundary Extraction from Radio-Echo Sounding Data

Marcel Dreier[1], Moritz Koch[2], Nora Gourmelon[1], Norbert Blindow[2], Daniel Steinhage[3], Fei Wu[1], Thorsten Seehaus[2], Matthias Braun[2], Andreas Maier[1], and Vincent Christlein[1]

[1]Department of Computer Science, Friedrich-Alexander-Universität Erlangen-Nürnberg, Erlangen, Germany.
[2]Institut für Geographie, Friedrich-Alexander-Universität Erlangen-Nürnberg, Erlangen, Germany.
[3]Alfred Wegener Institute for Polar and Marine Research, Bremerhaven, Germany.

**Correspondence:** Marcel Dreier (marcel.dreier@fau.de)

**Abstract.** The measurement of ice thickness is of great importance for the accurate estimation of glacier volume and the delineation of their bedrock topography. In particular, this is a crucial factor in forecasting the future evolution of glaciers in the context of a changing climate. In order to derive the ice thickness, the travel time of electromagnetic waves in radargrams acquired by radio-echo sounding (RES) systems is analyzed. This can only be achieved by identifying the ice surface and underlying ice bottom in corresponding radargrams. Manually identifying these two reflection horizons in RES data is a laborious and time-consuming process. Consequently, scientists are attempting to automate this task through the use of techniques such as deep learning. Such automation can significantly reduce the time between a field campaign and the calculation of the glacier's ice thickness distribution. In this paper, we present the first benchmark dataset for delineating the ice surface and bottom boundaries in RES data, to facilitate straightforward comparisons of deep learning models in the future. The "IceAnatomy" dataset comprises radargrams and the corresponding manual picks, amounting to a total of over 45,000 km of observations. The RES data originates from three sources: FAU, CReSIS, and AWI. The dataset comprises different RES systems as well as different pre-processing methods. In addition, the data was acquired over a large range of geographical and glaciological settings, featuring different thermal regimes present in Antarctica and the Southern Patagonian Icefield. This diversity ensures that the models' behaviors can be analyzed in different scenarios. We define a standardized train-test split for each source in the dataset. This allows us to introduce not only a baseline model trained on the entire training set (the "omni" model), but also three source-specific baseline models. The source-specific models are trained exclusively on the subset of the training data acquired by the specified source. The baseline models provide an initial benchmark against which subsequent models can be compared. The source-specific models demonstrate more accurate results than the omni model. For the FAU, CReSIS, and AWI test sets, the source-specific models achieve Mean Meter Errors of $2.1\,\text{m}$, $23.1\,\text{m}$, and $4.9\,\text{m}$ for the ice surface and $9.1\,\text{m}$, $78.2\,\text{m}$, and $29.3\,\text{m}$ for the ice bottom. In relation to the mean measured ice thickness of the test set, these errors equate to $1.2\,\%$, $3.1\,\%$, and $0.3\,\%$ for the ice surface and $4.9\,\%$, $10.4\,\%$, and $1.5\,\%$ for the ice bottom. The dataset and implementation are available at https://zenodo.org/records/14036897 (Dreier et al., 2024) and https://doi.org/10.5281/zenodo.14038570 (Dreier, 2024).



## 1 Introduction

Glaciers and ice shelves are key indicators of global climate (Haeberli et al., 2007; IPCC, 2013). Knowing their volume and ice thickness distribution is crucial for assessing future cryospheric contributions to sea level rise. Moreover, data on the ice volume of glaciers and ice sheets is necessary for understanding their response to climate change. Ice thickness measurements enable the subsequent prediction of the rate and timing of glacier retreat or disappearance using different types of models. That way, a glacier's contribution to regional hydrological cycles and subsequent influence on local to regional scales with

associated socioeconomic impacts can be assessed (Werder et al., 2020; Ayala et al., 2020; Farinotti et al., 2017). Several techniques to determine ice thickness exist, including seismic, gravitational, and magnetic methods, as well as radio-echo sounding (RES) (Bogorodsky et al., 2012; Kohler et al., 1997). While satellite gravimetry allows for a resolution in the range of kilometers, its spatial resolution does not allow for the interpretation of detailed subglacial features (Willen et al., 2024). Seismic measurements offer a high resolution, but widespread use in the Antarctic region is limited by high exploration costs or

logistical unfeasibility (An et al., 2023). For this reason, RES is preferred over other methods when an accurate assessment of a subglacial topography is of interest. After pre-processing the RES data, a cross-section, a so-called radargram of the glacier, becomes visible. Experts can then interpret the RES data by delineating the reflections of surfaces or internal glacial structures. Delineating the ice boundary, defined by the air-ice layer and ice-ground layer, is necessary to obtain the glacier's thickness at each point in the radargram. However, it is a laborious task, especially with large datasets (Sime et al., 2011). Several

automated and semi-automated approaches to delineate the layers have been developed (Fahnestock et al., 2001; Gifford et al., 2010; Freeman et al., 2010; Rahnemoonfar et al., 2017a, b; Kamangir et al., 2018; Rahnemoonfar et al., 2019; Cai et al., 2020, 2022; Liu-Schiaffini et al., 2022b). However, these approaches are not comparable as they have been evaluated on different datasets or a different train-test split of the same dataset. In this paper, we present a publicly available, ready-to-use benchmark dataset for ice thickness extraction. It is the first of its kind to be directly conjured for deep learning approaches,

with a pre-defined train-test split, human-annotated labels, and different recording systems. It comprises radargrams from Antarctica and Patagonia with polythermal, cold-based, or temperate thermal regimes. The dataset is intended for supervised training and evaluation of deep learning models. Therefore, the dataset includes depth labels for both the ice bottom and ice surface layer. Together with the dataset, we present a baseline model that delineates the ice boundary in a given radargram. The model is based on the U-Net architecture (Ronneberger et al., 2015) and serves as a reference and a starting point for future

improvements.

In summary, our contributions are as follows:

1. A novel benchmark dataset IceAnatomy for deep learning-based extraction of ice boundary from RES data is created.

2. A baseline deep learning model for the automatic delineation of ice bottom and ice layer is proposed.

3. A thorough evaluation of individual models and an omni-model is conducted on the dataset.

The work is structured as follows: Section 2 provides an overview of datasets and algorithms previously used for automatic ice boundary extraction. Subsequently, Sect. 3 gives insight into the recording and processing of the dataset as well as relevant





geographical and glaciological factors of the study sites. The baseline models are introduced in Sect. 4. An extensive evaluation of the baseline models and the benchmark dataset is presented in Sect. 5. Lastly, we summarize our research and draw conclusions in Sect. 6.

## 60  2   Related Works

Over the past decades, RES has been widely used in glaciology. A multitude of publications cover the extraction of ice boundary layers from RES data. In this section, we highlight related RES datasets and layer extraction approaches.

### 2.1   Datasets

RES data on glaciers and ice sheets is abundantly available. However, a large portion of the associated bedrock labels are inac-
curate, generated automatically, or unavailable. Hence, we focus our comparison on datasets that have been used to extract the ice boundary in previously published work and for which both radargrams and human-annotated labels are publicly available. These constraints significantly limit the number of related datasets.

The one RES system that has been used extensively to collect such data is the Multichannel Coherent Radar Depth Sounder versions 1-5 (MCoRDS) (Allen et al., 2012a), which was used, for example, in NASA's Operation IceBridge (OIB) program
on a McDonnell Douglas DC-8-72 jetliner (Shi et al., 2010a). The data acquired over Antarctica in 2009 are the most widely used (Crandall et al., 2012; Lee et al., 2014; Rahnemoonfar et al., 2017a, b; Berger et al., 2018; Kamangir et al., 2018). However, also data from different years (Kamangir et al., 2018; Mitchell et al., 2013; Cai et al., 2020; García et al., 2021a, b; Cai et al., 2022, 2019; Ghosh and Bovolo, 2022; García et al., 2023; Donini et al., 2022; Ilisei and Bruzzone, 2014, 2015) and other locations like Greenland (Donini et al., 2022) and the Canadian Arctic Archipelago (Xu et al., 2017, 2018) were
analyzed.

Only very few publications included data from RES systems other than MCoRDS. Gifford et al. (2010) extracted the ice boundary from data acquired by a predecessor RES system (Lohoefener, 2006) during 2006 and 2007 in Greenland. Dong et al. (2022) featured data from the Chinese Academy of Sciences' Deep Ice Radar acquired during the 29th Chinese Antarctic Scientific Expedition. Lastly, Liu-Schiaffini et al. (2022a) used algorithm-assisted human-labeled data acquired in the Cana-
dian Arctic and Antarctica by the University of Texas Institute for Geophysics' high-capability radar sounder (HiCARS). In conclusion, to the best of our knowledge, there is no comparable benchmark dataset for ice boundary extraction from radio-echo-sounding data.

### 2.2   Algorithms

RES has been employed to detect crevasses (Liu et al., 2020; Walker and Ray, 2019; Williams et al., 2012, 2014), the ice
boundary (Crandall et al., 2012; Lee et al., 2014; Rahnemoonfar et al., 2017a, b; Berger et al., 2018; Kamangir et al., 2018; Mitchell et al., 2013; Xu et al., 2017, 2018; Cai et al., 2022; Gifford et al., 2010; Dong et al., 2022; Liu-Schiaffini et al., 2022a), to segment subsurface structures (Cai et al., 2020, 2019; García et al., 2021a, b; Ghosh and Bovolo, 2022; García et al., 2023;



Donini et al., 2022; Ilisei and Bruzzone, 2014, 2015), and to track internal ice and snow layers (Crandall et al., 2012; Karlsson et al., 2013; Ibikunle et al., 2020; Rahnemoonfar et al., 2021; Varshney et al., 2020, 2021; Yari et al., 2019, 2020; Dong et al., 2022).


To obtain the ice boundary, one can either directly delineate the ice surface and bottom or first segment different regions such as ice, bedrock, and air and then extract the two layers during post-processing. Most studies (Crandall et al., 2012; Lee et al., 2014; Rahnemoonfar et al., 2017a, b; Berger et al., 2018; Kamangir et al., 2018; Mitchell et al., 2013; Xu et al., 2017, 2018; Cai et al., 2022; Gifford et al., 2010; Dong et al., 2022; Liu-Schiaffini et al., 2022a) to date prefer direct extraction. Fewer

studies (Cai et al., 2020, 2019; García et al., 2021a, b; Ghosh and Bovolo, 2022; García et al., 2023; Donini et al., 2022; Ilisei and Bruzzone, 2014, 2015) use the segmentation approach. The segmentation approach assigns a semantic class to each pixel in the radargram, from which the ice boundaries can be derived directly or after post-processing.

In terms of methodology, early studies mainly used classical image processing and machine learning techniques such as Hidden Markov Models (Crandall et al., 2012; Berger et al., 2018), Markov-Chain Monte Carlo (Lee et al., 2014), contour

detection (Rahnemoonfar et al., 2017a), the level set approach (Rahnemoonfar et al., 2017b; Mitchell et al., 2013), Markov Random Fields (Xu et al., 2017), edge-based and active contour methods (Gifford et al., 2010), Kullback-Leibler maps (Ilisei and Bruzzone, 2014), and Support Vector Machines (Ilisei and Bruzzone, 2015). After 2017, studies turned to Convolutional Neural Networks (CNNs) (Kamangir et al., 2018; Cai et al., 2020, 2019; García et al., 2021a; Cai et al., 2022; Donini et al., 2022; Dong et al., 2022; Liu-Schiaffini et al., 2022a; García et al., 2021b, 2023), combinations of CNNs and Recurrent Neural

Networks (RNNs) (Xu et al., 2018), and combinations of CNNs and Transformers (Ghosh and Bovolo, 2022).

In comparison, we rely on the U-Net architecture from (Ronneberger et al., 2015) to evaluate our newly created dataset. Furthermore, we integrate Atrous Spatial Pyramid Pooling from (Chen et al., 2018) and the ResBlock design from (Esser et al., 2020) to improve the performance.

## 3 Dataset

In this section, we introduce the benchmark dataset "IceAnatomy" which covers several different geolocations and was acquired by multiple radar systems. We divide the dataset into three subsets based on the sources of the data: the AWI (Alfred Wegener Institute, Helmholtz Centre for Polar and Marine Research), CReSIS (The Center for Remote Sensing and Integrated Systems), and FAU (Friedrich-Alexander-Universität Erlangen-Nürnberg, Institute of Geography) subsets. A summary of the most important information about the dataset is given in Tab. 1.

### 3.1 Study Sites

#### 3.1.1 Southern Patagonian Icefield

The Southern Patagonian Icefield (SPI) is the largest temperate ice body in the Southern Hemisphere. It is characterized by one of the highest mass loss rates in the world (Zemp et al., 2019; Marzeion et al., 2018; Hugonnet et al., 2021) and by





**Table 1.** A summary of details about the IceAnatomy benchmark dataset (Lippl et al., 2019; Shi et al., 2010b; Rückamp and Blindow, 2012; CReSIS, 2024a; Allen et al., 2012b; Steinhage, 2001, 2015).

| | Study Sites | Depth-Reso. | Width-Reso. | Length | Year | Main Thermal Regime | Labeled Bottom % |
|---|---|---|---|---|---|---|---|
| FAU | James Ross Island | $2.5\,\text{ns pixel}^{-1}$ | $2\,\text{m pixel}^{-1}$ | $275\,\text{km}$ | $2017/18$ | Polythermal | $82.5\,\%$ |
| | Perito Moreno | $2.5\,\text{ns pixel}^{-1}$ | $2\,\text{m pixel}^{-1}$ | $145\,\text{km}$ | $2022$ | Temperate | $83.1\,\%$ |
| | Viedma | $2.5\,\text{ns pixel}^{-1}$ | $2\,\text{m pixel}^{-1}$ | $140\,\text{km}$ | $2022$ | Temperate | $46.2\,\%$ |
| CReSIS | Antarctic Peninsula | $105\,\text{ns pixel}^{-1}$ | $12\,\text{m pixel}^{-1}$ | $20400\,\text{km}$ | $2009$ | Polythermal | $63.9\,\%$ |
| | West Antarctica | $105\,\text{ns pixel}^{-1}$ | $12-30\,\text{m pixel}^{-1}$ | $24400\,\text{km}$ | $2009$ | Polythermal | $78.9\,\%$ |
| AWI | Antarctic Peninsula | $12\,\text{ns pixel}^{-1}$ | $62\,\text{m pixel}^{-1}$ | $1490\,\text{km}$ | $2013$ | Polythermal | $31.7\,\%$ |
| | East Antarctica | $13.33\,\text{ns pixel}^{-1}$ | $66-79\,\text{m pixel}^{-1}$ | $1015\,\text{km}$ | $1997/99$ | Cold-based | $73.7\,\%$ |

its large outlet glaciers that drain into lakes or the ocean (Aniya, 1999). Two of the largest eastward-flowing outlet glaciers
in the region are the Perito Moreno and Viedma glaciers. The only way to obtain information over large areas about their
bedrock topography is by helicopter-borne RES measurements. This is particularly applicable to the lower parts of the glaciers,
which are surrounded by steep mountain flanks and have heavily crevassed surfaces. The temperate nature of the glaciers,
resulting in high water content in the ice, combined with the steep and deep glacier troughs, often makes analyzing radargrams
challenging. Hence, they pose an adequate challenge to benchmark new machine learning systems. Figure 1 shows the location
of both glaciers on the east of the SPI.

### 3.1.2 Antarctica

As depicted in Fig. 2, the Ice-Anatomy dataset offers three major study sites in Antarctica: the Antarctic Peninsula (including
James Ross Island (JRI)), West Antarctica, and East Antarctica.

The Antarctic Peninsula is the most represented region in the benchmark dataset, as it is present in all three RES subsets. It
exhibits one of the milder climates in Antarctica, with an annual average temperature of $-3.2\,°C$ (Morris and Vaughan, 1994).
This is also reflected in the thermal regimes present in the region, as it contains temperate, cold-based, and polythermal ice. The
temperate portions of the Antarctic Peninsula are frequently near the margins and at lower elevations, while the cold-based ice
regions are generally found at higher elevations. The transition zones between higher and lower elevations commonly contain
polythermal ice (Van Liefferinge and Pattyn, 2013; Macelloni et al., 2019). However, elevation alone is often not sufficient to
determine the thermal regime. If we compare the ice velocity maps of Rignot et al. (2011), we observe fast-moving ice even in
higher elevation areas, which is atypical for cold-based areas. This leads us to the assumption that there is a significant amount
of polythermal ice at higher elevations and that the main thermal regime is polythermal. Another noteworthy characteristic
of the Antarctica Peninsula is its relatively shallow ice sheet compared to the rest of Antarctica. On average, the ice sheet is
estimated to be $610\,\text{m}$ thick inland and $300\,\text{m}$ in the ice shelves (Drewry et al., 1982). This results in a generally clearer signal



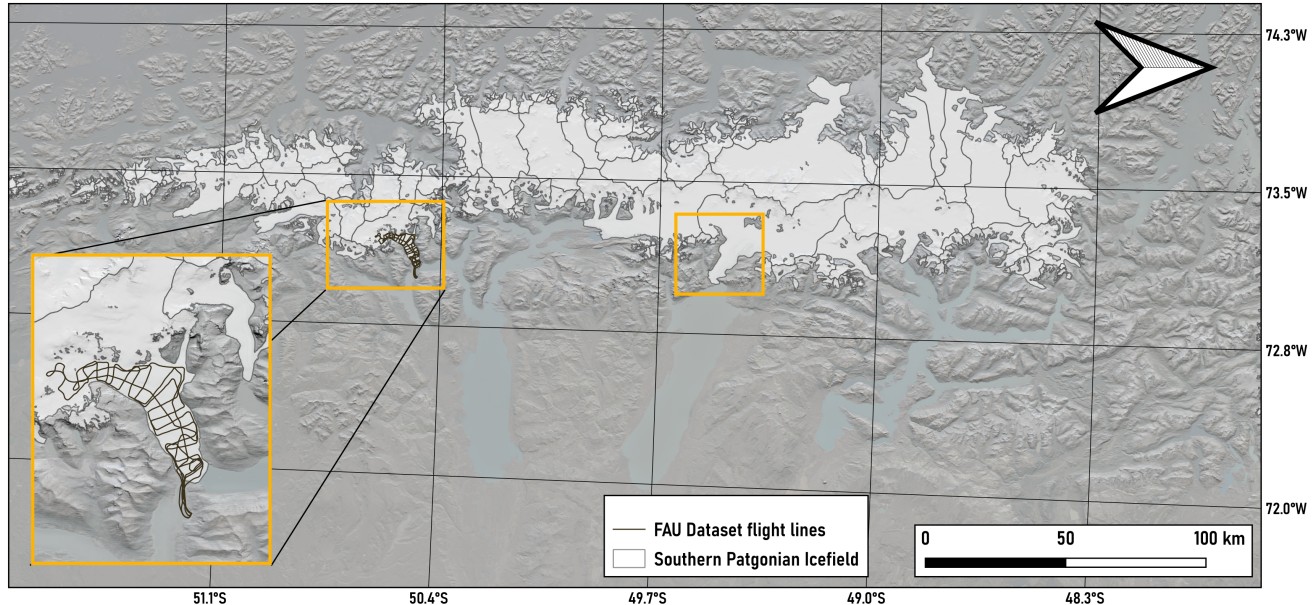

**Figure 1.** Overview of the Southern Patagonian Inland Icefield. Orange boxes indicate surveyed areas of Perito Moreno Glacier and Viedma Glacier. Black lines indicate flight paths over the Perito Moreno Glacier. The background is a hillshaded SRTM over ©Google Earth optical imagery (Consortium, 2017). Maps are rotated by 90 degrees.

because the signal has to travel through less ice. Hence, the returned amplitudes are usually higher and are less likely to be distorted by impurities in the glacier.

West Antarctica is significantly colder than the Antarctic Peninsula, with an annual average temperature of approximately $-28.1\,^{\circ}C$ and a primarily polythermal thermal regime (Morris and Vaughan, 1994). Polythermal regions only commonly occur at the margins and the coastline, while cold-based zones are mainly present at higher elevations (Macelloni et al., 2019;

Van Liefferinge and Pattyn, 2013; Rignot et al., 2011). West Antarctica also contains relatively thick ice with inland ice sheets estimated to be 1780 m thick and ice shelves around 375 m (Drewry et al., 1982).

The last subregion in Antarctica is East Antarctica. It exhibits the coldest climate of the three areas, with an annual average temperature of around $-59.8\,^{\circ}C$ and a primarily cold-based thermal regime (Morris and Vaughan, 1994). Temperate areas are commonly only near the margins, while polythermal zones act as a transition (Macelloni et al., 2019; Van Liefferinge and

Pattyn, 2013; Rignot et al., 2011). East Antarctica is also the region with the generally thickest ice. On average, its ice sheets are approximately 2630 m thick inland and 400 m in its ice shelves (Drewry et al., 1982).





**Figure 2.** Flight paths of the AWI and CReSIS campaigns in Antarctica. The background is assembled with help from the Quantarctica QGIS project (Matsuoka et al., 2021).

## 3.2 Dataset Generation

### 3.2.1 FAU Data

The RES system of FAU is a broadband 25 MHz bi-static monopulse sounder designed as a sling load for helicopter use. It is
a functional clone of the BGR-P30 system (Blindow et al., 2012). The antenna weighs roughly 280 kg and can be attached to any helicopter type that allows for the attachment of a sling load and has the required take-off capacity. The system is typically operated 20 m above ground at a nominal airspeed of 60 km h$^{-1}$.

The radar time series are collected at a 2.5 ns sampling rate using 256-fold stacking to improve sensitivity and signal-to-noise ratio. The traces are collected at a rate of 10 Hz, corresponding to approximately a 2-meter spatial sampling rate. The data are
georeferenced by two Leica GS16 multifrequency Global Navigation Satellite System (GNSS) systems. The rover antenna is



mounted on the radar antenna in a central position, while the base station is installed in proximity to the landing and starting area. After differential processing of the GNSS data, the positions are matched to the radar traces before further processing is applied. Then, the RES data is processed in REFLEX v8.5 software, developed by Sandmeier Geophysical Research. The processing flow comprises the following steps and is applied to subsections of each flight: equidistant trace interpolation,

shift for time zero, subtracting special average, bandpass filter, amplitude regulation by gain function (cold ice) or energy decay (temperate ice), 2D migration, static correction. To apply the 2D migration, it is necessary to derive a velocity model comprising an air and an ice layer. For the air layer, the wave travels at the speed of light, while for the ice layer, we assumed a speed of $0.168 \, \mathrm{m \, ns^{-1}}$ (Johari and Charette, 1975). Especially in temperate ice, the migration helps to focus the scattered energy to enhance the bedrock reflections.

The RES data of JRI was acquired during two different airborne ground penetrating radar campaigns in 2017 and 2018 (Lippl et al., 2019). Since Gourdon Glacier consists mainly of bare ice, no firn correction was applied for the outer parts of the profile. For the data on the plateau, a standard correction for firn and snow (+10 m, AWI/BAS Bedmap 1 mission summary) as used in the British-Argentinian survey was assumed (Lythe and Vaughan, 2001). The RES data of Perito Moreno Glacier and Viedma Glacier were acquired in March and April 2022. For these study sites in the FAU subset, the radargrams go deeper than the

radar's maximum penetration depth of 700 m (Blindow et al., 2011). The original depth of the radargrams is over 6000 pixels, which equates to over 1300 m on average - the total depth in meters is not constant due to fluctuations in the flight height of the helicopter. We cut the radargrams to 4096 pixels, which corresponds to an average of about 800 m. This way, we save computing power while keeping all the necessary information. To restore the full flight traces in the FAU dataset from their subsampled parts, we reassembled the radargrams according to their trace numbers. Any conflicting depths for the ice surface

and bottom in overlapping parts were smoothened with Gaussian importance weighting. Furthermore, we filled gaps of eleven pixels or less in the ice surface and bottom via bicubic interpolation.

### 3.2.2 CReSIS Data

The CReSIS data was recorded during the 2009 campaign of Operation Ice Bridge in Antarctica, which comprised 21 missions. Three were sea-ice surveys and thus are not included in the CReSIS dataset. The remaining 18 missions can be split into

two groups: six missions focusing on the Antarctic Peninsula (PEN1, PEN2, PEN3, PEN4, PEN5, and LVISPEN) and 12 missions exploring West Antarctica (PIG1, PIG2, PIG3, PIG4, LVISPIG, LVIS86, GETZ1, ABBOTT1, TSK1, TSK2, TSK3, and TSK4) (Allen et al., 2012b). All 18 missions employed the Multichannel Coherent Radar Depth Sounder (MCoRDS) flown on a McDonnell Douglas DC-8-72. It has a center frequency of 195 MHz and an eight-channel-chirp signal to accurately assess the ground (Rodriguez-Morales et al., 2014; Shi et al., 2010b).

To process the recorded data, the standard CReSIS L1B CSARP-mvdr (minimum variance distortionless response) processing steps were applied. These include pulse compression via a Tukey and Hanning Window, beam-forming, motion compensation, synthetic aperture radar processing in combination with f-k migration, channel combination, and waveform combination (CReSIS, 2024b). After the processing, the radargrams had a depth resolution of $105 \, \mathrm{ns \, pixel^{-1}}$ and a width resolution of $12 - 30 \, \mathrm{m \, pixel^{-1}}$ depending on the mission.





We obtained the fully processed CReSIS subset by downloading the CSARP-mvdr processed L1B product from the CReSIS website and taking the square root of the amplitudes. Likewise, CReSIS also provides downloads for the annotated glacier and bedrock surface layers on their website (CReSIS, 2024a). According to Lee et al. (2014) and Crandall et al. (2012), the rock-bed surface is humanly annotated but noisy. Although the noise might pose a problem for certain approaches, we chose not to alter the labels. The reason for this is that the dataset has been used previously in other publications, and in order to

remain comparable, we use the same labels. Nonetheless, we also include the CReSIS quality labels in the benchmark dataset to highlight the picker's confidence in each label.

### 3.2.3 AWI Data

The AWI subset was recorded during campaigns in Dronning Maud Land in 1997 and 1999 (Steinhage et al., 2023b, a) and in the Antarctic Peninsula in November 2013 (Steinhage, 2015). All three campaigns employed a version of the EMR radar

system with a center frequency of 150 MHz and the toggle mode enabled. The toggle mode alternates the radar's pulse length between 60 ns and 600 ns periodically. Thus, the system can achieve a decent depth resolution while capturing deep internal layers of the ice. The processing of the recorded data was similar for all three campaigns. The data was differentiated, rescaled, high-pass filtered, and bandpass filtered. To reduce the amount of noise in a radargram, multiple traces were combined into a single trace. In detail, ten traces were combined for the 1997 and 1999 flights, and seven traces were combined for the

2013 flight (Steinhage, 2001; Nixdorf et al., 1999; Steinhage et al., 2001). Automatic gain control was used to normalize the amplitude values. After the processing, the radargrams had a depth resolution of $12 - 13.33\,\mathrm{ns\,pixel^{-1}}$ and a width resolution of $66 - 79\,\mathrm{m\,pixel^{-1}}$ depending on the campaign.

    In the picks, gaps of eleven pixels or less were filled using bicubic interpolation. Finally, for the radargrams from 1997 and 1999, everything below 3600 pixels, which is about 4 km, was discarded because only noise was visible at these depths. The

gathered data was processed with FOCUS, DISCO, LANDMARK, and Python.

## 4   Baseline Method

To demonstrate the usability of the dataset, we present a baseline method in this section. The method's pipeline consists of preprocessing steps and a deep learning model, elaborated in the following subsections.

### 4.1   Preprocessing

The radargrams are given in relative power $p$ to the recorded amplitudes, which we first convert to decibels using the following formula:

$$dB = 10 \cdot \log_{10}(p) \tag{1}$$

Next, we apply a z-score normalization, i.e., we subtract the mean and divide by the standard deviation. However, the mean and standard deviation are not formed over the entire IceAnatomy dataset because there is a strong divergence in the recorded



spectrum values between the different subsets. This divergence is caused by the large difference in radar systems and processing of the data, which represents a domain shift. Therefore, the normalization is performed separately for the AWI and CReSIS data and for the three study sites in the FAU subset.

Then, the normalized radargrams of the entire IceAnatomy dataset are resized to a standard height of 1024 pixels. Finally, each radargram is cut into patches with a width of 512 pixels and a total height of 1024 pixels. For trajectories whose width is 230 not divisible by 512, we apply symmetric padding at the end.

## 4.2 Deep Learning Model

We apply a deep learning model to extract the ice boundary from the radargram. The model's architecture is depicted in Fig. 3 and is based on the U-Net (Ronneberger et al., 2015).

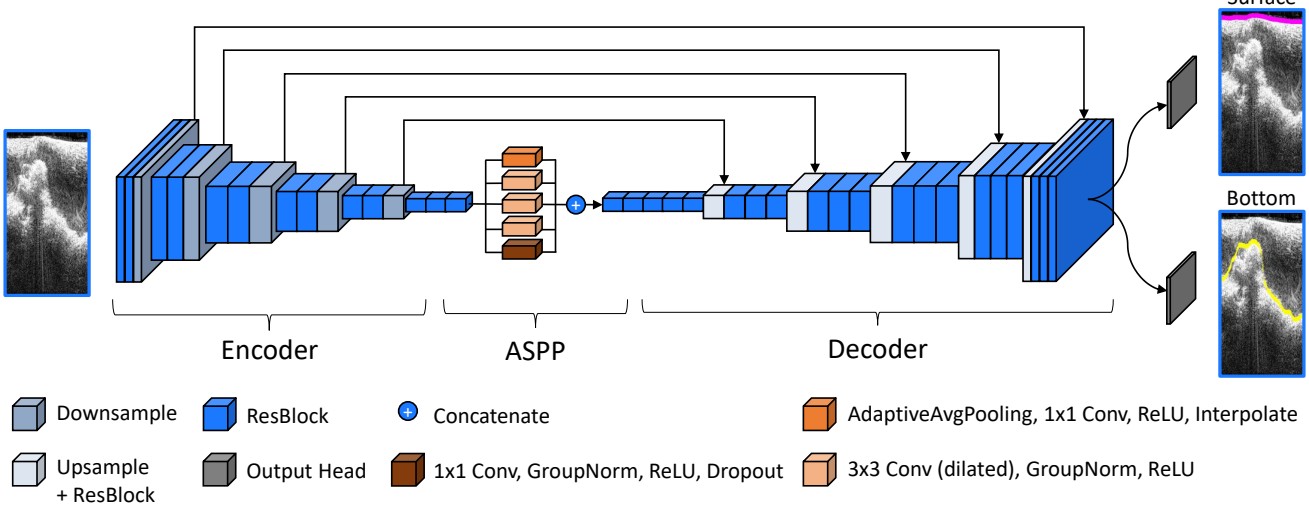

**Figure 3.** The architecture of the proposed deep learning model. It receives the normalized amplitudes of a radargram as input and predicts the glacier surface and the bedrock surface as two separate outputs. The atrous spatial pyramid pooling contains three dilated convolutional layers, one convolutional layer with adaptive average pooling, and a $1 \times 1$ convolutional layer. It utilizes the rectified linear unit (ReLU) activation function which is defined as $ReLU(x) = max(x, 0)$.

It consists of three components: an encoder, a decoder, and a bottleneck. The encoder extracts features from the radargram 235 into a feature map, the decoder utilizes the feature map to make a prediction, and the bottleneck connects these two components. As the model has to handle large input sizes, the encoder contains five down-sampling steps to process the input, while the decoder has five up-sampling steps to reconstruct the original size. In the encoder, each down-sampling step consists of two residual blocks (ResBlocks), while in the decoder, each up-sampling step consists of three ResBlocks (Esser et al., 2020).

The structure of the ResBlock, shown in Fig. 4, comprises several key components. First, it starts with a group normalization 240 layer (Wu and He, 2018) that normalizes the data in groups of channels to increase stability during training. Next, a swish



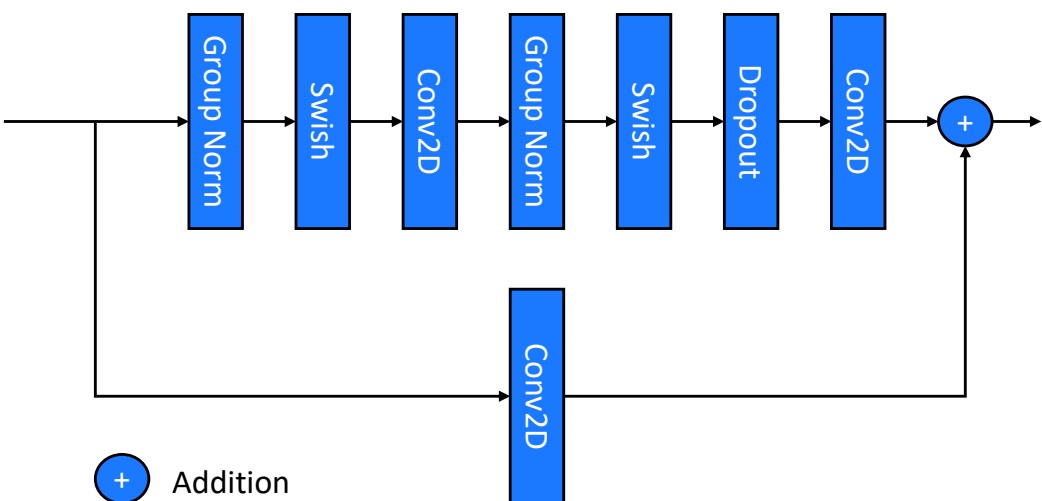

**Figure 4.** Structure of the residual block employed in our deep learning model. The arrangement is based on the design of Esser et al. (2020)

activation (Prajit Ramachandran, 2018) function adds nonlinearity to the ResBlock so the network can learn more complex patterns. The activation is followed by a two-dimensional convolution layer that processes and combines the visual features by applying convolutional operations. This is followed by another group normalization and swish activation function before a dropout layer (Hinton et al., 2012) is applied. The dropout layer randomly withholds information during training to improve

generalization and prevent the model from overfitting – a process in which the model develops a strong bias towards the training data. After the dropout layer, another two-dimensional convolutional layer is applied. Finally, a residual connection (He et al., 2016), a shortcut from the start of the ResBlock to the end through a convolution layer, is added to the output of this sequence of layers to improve the gradient flow in the network. In the bottleneck, we inserted an Atrous Spatial Pyramid Pooling (ASPP) (Chen et al., 2018) layer. ASPP processes the same feature map in parallel with differently dilated convolutional

layers. In contrast to typical convolutional layers, dilated convolutional layers do not utilize a set of adjacent pixels. Instead, they sample a set of pixels from a grid around a center point, thereby achieving differently sized fields of view. The sampling is uniform and based on a dilation rate. The chosen dilation rates in this model are 1, 4, and 6. Since the model is based on the U-Net architecture, it also includes skip connections. Skip connections directly forward the output of each down-sample step in the encoder to the corresponding parts in the decoder via concatenation. The increased channel dimensions in the decoder

are solved by including an additional ResBlock for channel reduction after each up-sampling step in the decoder.

To calculate the final prediction of the model, we first forward the feature map computed by the U-Net into two separate output heads, each consisting of a single ResBlock. Each output head then creates one probability map, resulting in two final probability maps. The first one represents the probabilistic prediction of the ice surface, while the second one represents the probabilistic prediction of the ice bottom. The final prediction of the model is then the highest probable prediction of each

column, which we compute by applying a column-wise argmax-operation.



To train the network, we employ a custom loss, a cost function that gives feedback to the network by measuring the difference between the prediction and the corresponding labeled ice boundary. The custom loss consists of two parts: a distance-based ($L_{\text{dist}}$) loss and a classification ($L_{\text{class}}$) loss:

$$L = L_{\text{dist}} + L_{\text{class}} \tag{2}$$

For the classification and distance-based losses, the probability maps of the ice surface ($\hat{Y}_{\text{s}}$) and ice bottom layer ($\hat{Y}_{\text{b}}$) are treated column-wise, i.e., per trace ($\hat{y}_{\text{s/b}}$). The classification loss is a smoothed cross-entropy loss ($L_{\text{CE}}$) that considers each pixel in a column $C$ as a separate class $c$. The pixel closest to the corresponding labeled ice boundary layer is then the correct class. The loss gets smoothed with the smoothing factor $\epsilon_c$. The distance-based loss sums up the probabilities in the column which are weighted with a distance map. The distance map contains the distance to the correct pick for each pixel. Hence, the further away the predicted pick is from the annotated layer, the greater the loss. The formulas of the classification and distance-based loss are as follows:

$$L_{\text{CE}} = -\sum_{c \in C} x_c(1 - \epsilon_c)\log(p(x_c)) + \frac{\epsilon_c(1 - x_c)\log(p(x_c))}{|C|} \tag{3}$$

$$L_{\text{class}} = \frac{w_{\text{s\_class}}}{|\hat{Y}_{\text{s}}|} \sum_{\hat{y}_{\text{s}} \in \hat{Y}_{\text{s}}} L_{\text{CE}}(\hat{y}_{\text{s}}) + \frac{w_{\text{b\_class}}}{|\hat{Y}_{\text{b}}|} \sum_{\hat{y}_{\text{b}} \in \hat{Y}_{\text{b}}} L_{\text{CE}}(\hat{y}_{\text{b}}) \tag{4}$$

$$L_{\text{dist}} = \frac{w_{\text{s\_dist}}}{|\hat{Y}_{\text{s}}|} \sum_{\hat{y}_{\text{s}} \in \hat{Y}_{\text{s}}} \langle d(y_{\text{s}}), \sigma(\hat{y}_{\text{s}}) \rangle + \frac{w_{\text{b\_dist}}}{|\hat{Y}_{\text{b}}|} \sum_{\hat{y}_{\text{b}} \in \hat{Y}_{\text{b}}} \langle d(y_{\text{b}}), \sigma(\hat{y}_{\text{b}}) \rangle \tag{5}$$

with $w_{\text{s\_class}}$, $w_{\text{b\_class}}$, $w_{\text{s\_dist}}$, and $w_{\text{b\_dist}}$ the respective weights for a weighted combination of the single loss parts. $\langle \rangle$ is the dot product, $\sigma$ is the softmax function that converts the model's outputs into probabilities, $|.|$ is the cardinality of a set, and $d$ is the function that creates a vector filled with the column-wise distance map given the respective column of the label.

The annotations in the dataset have discontinuities in the labeled layers where the ice bottom dropped below the radar's penetration depth, the receiver flew over the edge of the glacier, or the signal was too ambiguous for experts to interpret. Tracks for which no pick is available for a layer are not included in the loss calculation and the evaluation.

## 5 Evaluation

### 5.1 Evaluation Metrics

Previous work either directly extracted the ice boundaries or deduced them from an intermediate segmentation, where they predicted a semantic class for every pixel in the radargram. Depending on the chosen method, the metrics used to assess the quality of the predictions differ. For segmentation approaches, most of these metrics are based on a confusion matrix that measures how accurately the model distinguishes between a chosen positive class and all the other classes, dubbed negative class. A confusion matrix contains four measurements: true positives (TP) (the number of correctly predicted pixels for the positive class), true negatives (TN) (the number of correctly predicted pixels for the negative class), false positives (FP) (the



number of wrongly predicted pixels for the positive class), and false negatives (FN) (the number of wrongly predicted pixels for
the negative class). Based on these four measurements, more sophisticated metrics are defined for the segmentation approaches.
The most commonly employed one is the accuracy $\left(\frac{TP+TN}{TP+FP+FN+TN}\right)$ (García et al., 2021a, b, 2023; Ghosh and Bovolo,
2022; Donini et al., 2022; Ilisei and Bruzzone, 2015). Less commonly used metrics include the Intersection over Union (IoU)
$\left(\frac{TP}{TP+FP+FN}\right)$ (Cai et al., 2019), precision $\left(\frac{TP}{TP+FP}\right)$ (Ghosh and Bovolo, 2022), recall $\left(\frac{TP}{TP+FN}\right)$ (Ghosh and Bovolo, 2022),
the F1-score $\left(2\frac{Precision\cdot Recall}{Precision+Recall}\right)$ (Cai et al., 2020; Ghosh and Bovolo, 2022), sensitivity $\left(\frac{TP}{TP+FN}\right)$ (García et al., 2023; Donini
et al., 2022), specificity $\left(\frac{TN}{TN+FP}\right)$ (García et al., 2023; Donini et al., 2022), and the error rate $\left(\frac{FP+FN}{TN+TP+FP+FN}\right)$ (Ilisei and
Bruzzone, 2014).

For direct extraction approaches, the mean column-wise absolute error also called mean absolute error (MAE) (Crandall
et al., 2012; Lee et al., 2014; Rahnemoonfar et al., 2017a; Berger et al., 2018; Mitchell et al., 2013; Xu et al., 2017, 2018;
Gifford et al., 2010; Dong et al., 2022; Liu-Schiaffini et al., 2022a) is the most common metric. It measures the average pixel-
wise distance between the annotated layer and the prediction. Other distance-based metrics include the median of the column-
wise mean absolute error (Lee et al., 2014; Rahnemoonfar et al., 2017a; Berger et al., 2018; Xu et al., 2017), the mean squared
error (MSE) (Crandall et al., 2012; Mitchell et al., 2013; Dong et al., 2022), the root mean square error (RMSE) (Liu-Schiaffini
et al., 2022a), and the largest under- and over-estimation (Gifford et al., 2010). One problem with confusion matrix-based
metrics like the precision is that they are not distance-weighted. For example, if a prediction is always one pixel next to the
annotated layer, also known as ground truth (GT), the confusion matrix-based metrics will have the worst possible value, even
though it is a near-perfect prediction. Therefore, some studies (Xu et al., 2017; Gifford et al., 2010; Liu-Schiaffini et al., 2022a)
have relaxed these confusion matrix-based metrics by considering predictions a few pixels from the ground truth as still correct.

As metrics for our benchmark framework, we have chosen the MAE, two relaxed Average Precision (AP) metrics, and
introduce the Mean Meter Error (MME). The MAE is calculated as the column-wise difference in pixels between the ground
truth depth of a layer and the predicted depth. Resizing the radargram will change the value of this metric. Therefore, we also
introduce the MME, which approximates the real-world error. We calculate the MME by multiplying the MAE with the product
of the wave velocity in the medium and the depth resolution of the radargram. The speed of the wave in the medium is assumed
to be constant with the speed of light ($c_{air} = 0.299792458 \text{ m ns}^{-1}$) for the air layer and $c_{ice} = 0.168 \text{ m ns}^{-1}$ for the ice layer. The
depth resolution is the time it takes for the wave to pass through a pixel. Since the depth resolution is indirectly proportional
to the y-dimension of the radargram, the MME stays consistent across different heights. Table 1 records the different depth
resolutions for radargrams in the IceAnatomy dataset in their original height and equation 6 and 7 summarize the formula for
the MME.

$$\text{MME}_s(\hat{y}_s, y_s) = \frac{c_{air}}{2|\hat{Y}_s|} \sum_{\hat{y}_s \in \hat{Y}_s} \text{Depth-Reso}^* \cdot \text{MAE}_s(\hat{y}_s, y_s) \tag{6}$$

$$\text{MME}_b(\hat{y}_b, y_b) = \frac{c_{ice}}{2|\hat{Y}_b|} \sum_{\hat{y}_b \in \hat{Y}_b} \text{Depth-Reso}^* \cdot \text{MAE}_b(\hat{y}_b, y_b) \tag{7}$$





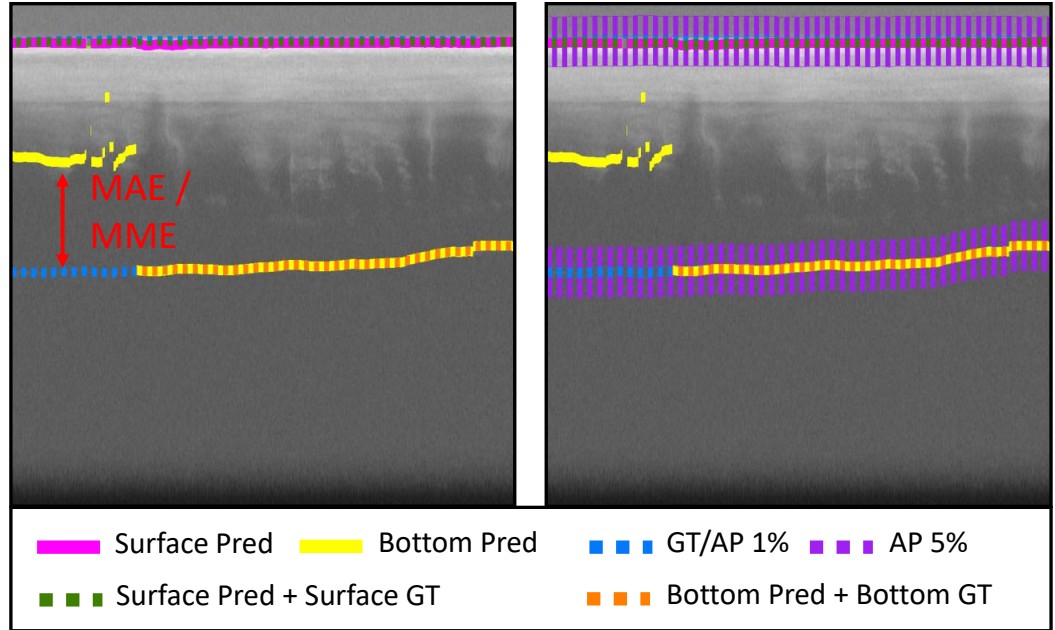

**Figure 5.** A visual representation of the four metrics used in this work. The left side of the figure depicts the MAE and MME respectively as the difference between prediction and ground truth (GT). Meanwhile, the right side of the figure features the AP-1% and AP-5% respectively as an interval around the ground truth. Note that the ground truth and the predictions are technically float numbers. However, we thickened the ground truth by 20 pixels to improve visibility.

(* Depth-Resolution after resizing the radargram.)

MME and MAE both describe the distance between two lines. A disadvantage is that they are not robust to outliers. As an outlier robust alternative, we also use a relaxed average precision (AP). To normalize the relaxation, we count everything below a 1 or 5 % error of the total height in pixels of the radargram as a hit (AP-1% and AP-5%). Choosing a relative error bound instead of fixing it to an absolute pixel value prevents the metric from changing when the radargram is resized. In addition, relaxing the metric alleviates the problem of uncertainties in the labels. Figure 5 shows a visualization of the employed metrics.

### 5.2 Experimental Protocol

Since there are large differences between the subsets of the IceAnatomy dataset, we train one model for each subset, i.e., the FAU, CReSIS, and AWI subsets. The model for the AWI data is a special case, as the subset is very small. This would make the model prone to overfitting. To counteract that, the AWI model is first pre-trained on all three subsets of the IceAnatomy dataset and then finetuned on the AWI subset. In addition to the specialized models, we train one model on the full IceAnatomy training dataset and evaluate it on the test subsets separately to contrast it to the subset models.



For the FAU subset, we select one flight from each of the study sites as part of the test set: The third flight over Perito Moreno, the second flight over Viedma, and the flights from 2017 for JRI. The remaining flights are used for training and validation, where the validation set includes the second half of the first flight over Perito Moreno, the third section of the first flight over JRI, and the traces 5023 to 8077 for the flight over Viedma. For the CReSIS subset, we choose the TSK2, PIG4, PEN4, and PEN5 missions as the test set. This results in 7 flights in the test set, containing 3 over the Antarctic Peninsula and 4 over West Antarctica. From the remaining 25 flights, the flights from PEN3, PIG3, and GETZ1 missions are taken for the validation set. For the AWI subset, we decided not to pick an exclusive flight for testing as the differences between the collected radargrams are too big. Instead, we utilized the last 20 % of the 2014 flight over the Antarctic Peninsula and the 1999 flight over East Antarctica as our test set. For training, we picked the entirety of the 1997 flight over East Antarctica, the first 70 % of the flight over the Antarctic Peninsula, and the first 70 % of the 1999 flight over East Antarctica. The remaining 10 % of the 1999 and 2014 flights were used for validation.

We assess the model on the validation set after every iteration over the full training set and stop training when the AP-1% does not improve for 25 subsequent evaluations. We save the model with the highest AP-1% value on the validation set. The learning rate, a parameter that determines the strength of every network update, is set to $5e^{-4}$. As the optimizer, an algorithm that updates the network weights based on the loss function, we use AdamW (Loshchilov and Hutter, 2019) with a weight decay of $0.05$ and reduce the learning rate by a factor of $0.5$ when AP-1% plateaus for ten subsequent iterations of the entire validation set. The batch size, a parameter that determines how many samples are used for every weight update, is 32 for all models. To increase variety in the data, we randomly modify the training data via data augmentations. In particular, we employ an additive Poisson noise scaled with Gaussian noise, brightness, contrast, gamma correction, and flipping horizontally.

## 5.3 Results

Table 2 provides quantitative results on all three subsets for the dataset-specific models and the omni model trained on the full dataset. Overall, the results are promising, with high AP-1% and AP-5% values and low MME and MAE values for most combinations. Still, dataset and model-specific discrepancies exist.

### 5.3.1 Ice Surface Predictions

The predictions for the ice surfaces are nearly perfect for all subsets and all models. The three subset models even achieve 100 % accuracy for the AP-5%. Hence, the remaining discrepancies are likely significantly influenced by measurement inaccuracies, noise, and general model variance. Therefore, we will only consider the task of ice bottom delineation to assess model performance.

### 5.3.2 Ice Bottom Predictions

For the ice bottom predictions, the differences in the MME between the three subsets are more pronounced than for the MAE, which can be attributed to the different depth resolutions. The MAE difference between the FAU and CReSIS subsets is small,





**Table 2.** Overview of the performance of our presented deep learning model on the different subsets in our benchmark dataset. We distinguish the layer prediction into two classes: the ice surface (S) and the ice bottom (B). Furthermore, we split our experiments into two parts: The dataset specific models, which were trained only on a specific subset of the data, and the omni model, which was trained on the entire dataset. Note that for the AWI subset-specific model, we utilized the weights of the omni model as a starting point to stabilize training. We compare the model's performance on the MME, MAE, AP-1%, and AP-5% as defined in Section 5.1. To contextualize the MME, we annotate the relative error to the mean measured ice thickness of the specified test set study site behind the MME. We conducted the evaluation on the test set and averaged the results over five runs to minimize statistical errors.

| | | Dataset specific Model | | | | Omni Model | | | |
|---|---|---|---|---|---|---|---|---|---|
| | Layer | MME ↓ | MAE ↓ | AP-1% ↑ | AP-5% ↑ | MME ↓ | MAE ↓ | AP-1% ↑ | AP-5% ↑ |
| FAU | S | 2.1 m [1.2 %] | 2.0 | 98.8 % | 100.0 % | 2.4 m [1.3 %] | 2.3 | 98.5 % | 99.9 % |
| | B | 9.1 m [4.9 %] | 13.1 | 74.3 % | 95.8 % | 19.5 m [10.5 %] | 27.3 | 68.3 % | 90.5 % |
| CReSIS | S | 23.1 m [3.1 %] | 2.5 | 96.9 % | 100.0 % | 20.8 m [2.8 %] | 2.2 | 97.9 % | 100.0 % |
| | B | 78.2 m [10.4 %] | 15.2 | 87.9 % | 94.1 % | 66.5 m [8.9 %] | 12.8 | 88.6 % | 94.4 % |
| AWI | S | 4.9 m [0.3 %] | 0.7 | 99.3 % | 100.0 % | 12.0 m [0.6 %] | 1.7 | 97.6 % | 99.4 % |
| | B | 29.3 m [1.5 %] | 7.4 | 83.5 % | 97.6 % | 39.8 m [2.1 %] | 10.0 | 75.7 % | 95.6 % |

while the MAE on the AWI subset is substantially lower than both. The AP-1% is lower for the FAU subset than for the AWI and CReSIS subsets. Interestingly, this difference between subsets is relativized for AP-5%. This means that most incorrect predictions for FAU are in the 1 % to 5 % error range. The same is true for the AWI subset. For the CReSIS data, this effect is not as strong. Here, the AP only increases from 87.9 % for the 1 % error rate to 94.1 % for the 5 % error rate.

### 5.3.3 Omni Model

The omni model shows persistently higher MME and MAE values and lower AP-1% and AP-5% values for the FAU and AWI subsets than the dataset-specific models. In detail, it only achieves an MME of 19.5 m and 39.8 m and an AP-1% of 68.3 % and 75.7 %, respectively. We attribute the lower performance of the omni model to the substantial domain shift between the three subsets and the fact that the FAU and AWI subsets are significantly smaller than the CReSIS subset. For the CReSIS subset, the omni model outperforms the dataset-specific model. In particular, it achieves an MME of 66.5 m and an AP-1% of 88.6 %. These results suggest that there can be a benefit from more training data even with the domain shift. However, the domain shift makes the generalization to under-represented or new domains difficult.

### 5.3.4 Influence of Study Sites

Table 3 divides the results of the subset-specific models by study site and thermal regime.

For the FAU subset, the Perito Moreno and Viedma predictions are quantitatively worse than the ones from JRI. A noticeable difference between Perito Moreno, Viedma, and JRI is the thermal regime. The first two are temperate glaciers, while JRI





**Table 3.** Overview of the influence of geographical and glaciological factors on the performance in detecting the ice bottom. We differentiate between the subset, the study site, and the general thermal regime. For the performance analysis, we compare the MME, MAE, AP-1%, and AP-5% as defined in Section 5.1. To contextualize the MME, we annotate the relative error to the mean measured ice thickness of the specified test set study site behind the MME. Note that for the AWI subset-specific model, we utilized the weights of the omni model as a starting point to stabilize training. We conducted the evaluation on the test set and averaged the results over five runs to minimize statistical errors. The analyzed models were the subset-specific models.

|  | **Study Site** | **Main Thermal Regime** | **MME** ↓ | **MAE** ↓ | **AP-1%** ↑ | **AP-5%** ↑ |
|---|---|---|---|---|---|---|
| FAU | Perito Moreno | Temperate | 22.1 m [8.0 %] | 26.3 | 54.9 % | 91.1 % |
|  | Viedma | Temperate | 10.0 m [5.0 %] | 12.0 | 68.5 % | 96.8 % |
|  | James Ross Island | Polythermal | 3.9 m [2.7 %] | 9.2 | 84.9 % | 96.9 % |
| CReSIS | Antarctic Peninsula | Polythermal | 31.6 m [4.5 %] | 5.8 | 91.5 % | 97.6 % |
|  | West Antarctica | Polythermal | 148.7 m [18.0 %] | 29.4 | 82.5 % | 88.8 % |
| AWI | Antarctic Peninsula | Polythermal | 32.7 m [9.8 %] | 8.1 | 87.3 % | 96.4 % |
|  | East Antarctica | Cold-based | 27.3 m [1.0 %] | 6.9 | 81.1 % | 98.3 % |

contains polythermal ice. Besides the higher water content in Perito Moreno and Viedma, both are also substantially deeper than JRI in most areas. They even exhibit areas with ice too thick for the employed radar system to penetrate. Viedma and JRI also feature several-meter-thick moraine material on the glacier surface. These rock and debris deposits are not penetrable by the wavelets and thus create radar shadows below them or substantially decrease the amount of reflected energy.

If we look at the associated radargrams, we can mostly see a relatively stable and clear prediction for JRI. On the other
hand, Viedma and Perito Moreno have much stronger differences to the ground truth. Especially in deep and noisy regions, the models struggle. Figure 6 shows example traces for the three study sites of the FAU subset.

Between the Antarctic Peninsula and West Antarctica study sites of the CReSIS subset, there are strong differences in the quantitative analysis. The MME and MAE values exhibit a difference of approximately a factor of five, while the AP-1% and AP-5% are approximately 9 % apart. In the qualitative analysis, we can see that the predictions in both regions actually
follow the ground truth closely. However, sometimes the predicted ice bottom layer makes a jump and the actual ice surface is predicted to be the ice bottom. We call this "ice boundary collapse". Examples of this phenomenon can be seen in Fig. 7.

For the AWI subset, the results for East Antarctica are more favorable than those for the Antarctic Peninsula, with the exception of the AP-1%. This outcome is in line with the observation on the FAU subset, that the algorithm and radar system performs better for colder ice performs as for warmer ice. It is noteworthy that the Antarctic Peninsula exhibits a superior
AP-1% compared to East Antarctica, yet a comparatively inferior AP-5%. This suggests that the inaccuracies inherent in the model's results for East Antarctica are not large, which is further supported by the lower MAE and MME. This phenomenon is particularly evident in the qualitative analysis, where the prediction aligns closely with the ground truth in East Antarctica. Similarly, the prediction for the Antarctic Peninsula also appears to be relatively accurate, although there are occasional outliers, which contribute to the elevated MME. Example traces can be seen in Fig. 8.



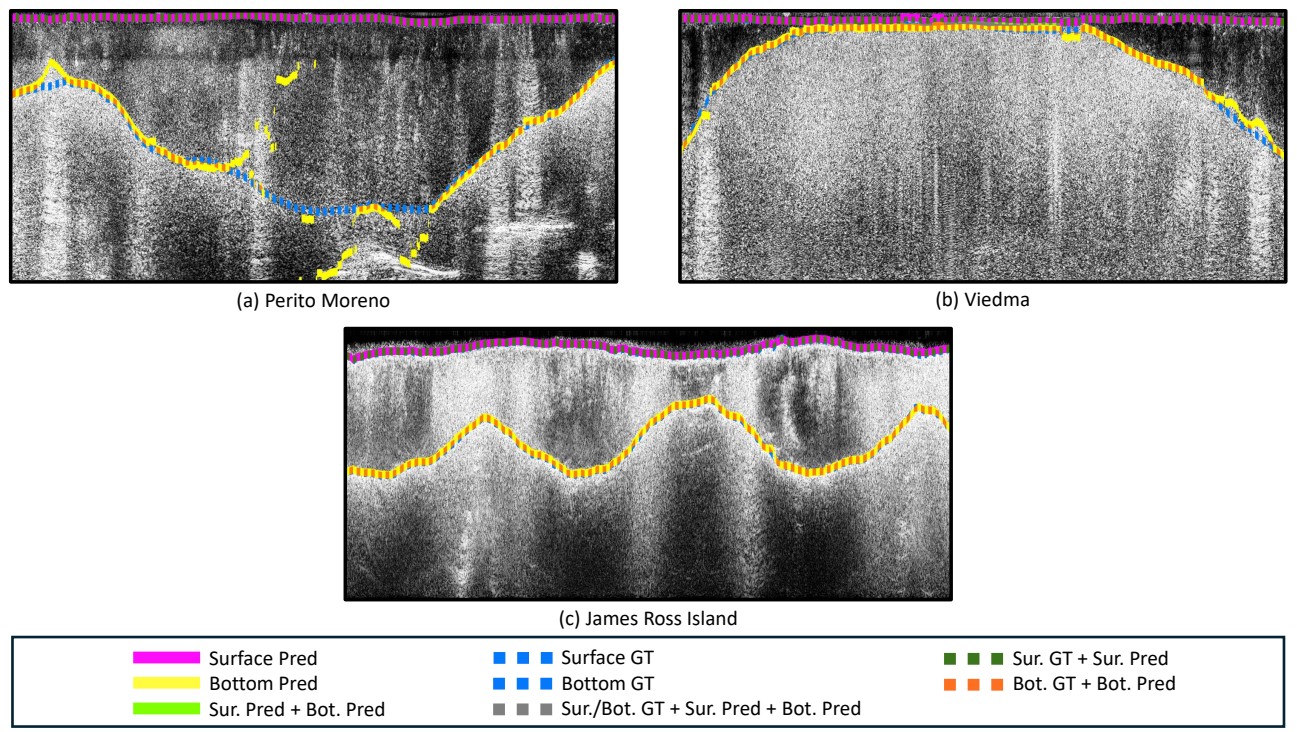

**Figure 6.** Visualization of the subset-specific model's performance on the FAU subset. Figure (a) shows trace 3000-5500 of the third flight over Perito Moreno, Fig. (b) depicts traces 5000-7500 of the second flight over Viedma, and Fig. (c) presents traces 5000-7500 from the first Section of the 2017 flights over James Ross Island.

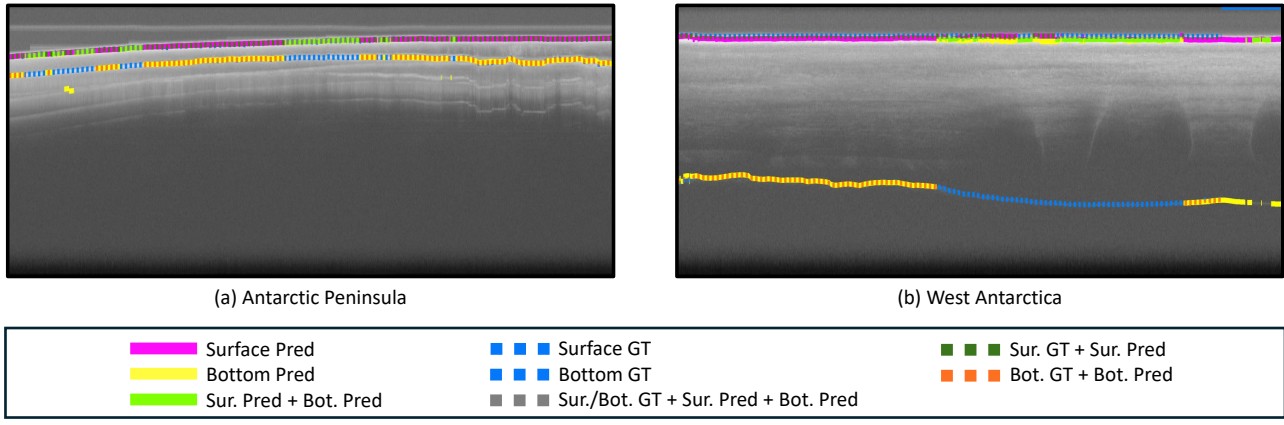

**Figure 7.** Visualization of the subset-specific model's performance on the CReSIS subset. Figure (a) presents traces 2000-4500 from mission PEN4 in the Antarctic Peninsula (PEN4_01_001). Figure (b) presents traces 2000-4500 from mission TSK2 in West Antarctica (TSK2_07_003).



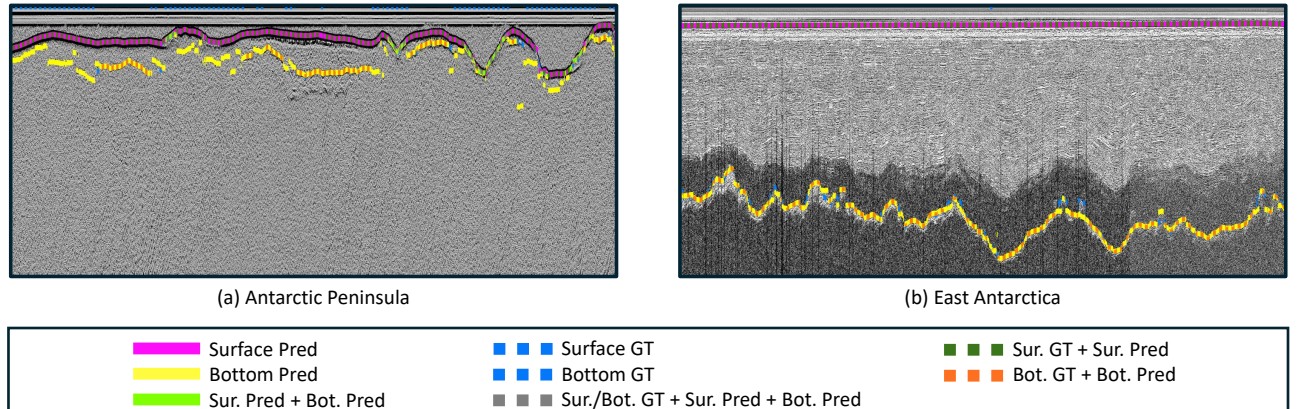

**Figure 8.** Visualization of the subset-specific model's performance on the AWI subset. Figure (a) depicts traces 21000-23500 from the 2014 flight in the Antarctic Peninsula. Figure (b) presents traces 7837-9787 from the 1999 flight in East Antarctica.

### 5.3.5 Loss Function

**Table 4.** Summary of our ablation study regarding the proposed modifications to the loss function. For every variation of the loss function, we trained a subset-specific model and compared the performance based on the MME and AP-1% of the ice bottom layer. We conducted the evaluation on the test set and averaged the results over five runs to minimize statistical errors. To contextualize the MME, we annotate the relative error to the mean measured ice thickness of the specified test set study site behind the MME. Note that for the AWI subset-specific model, we utilized the weights of the omni model as a starting point to stabilize training, which was also trained with the specified loss function.

| | FAU | | CReSIS | | AWI | |
|---|---|---|---|---|---|---|
| | MME ↓ | AP-1% ↑ | MME ↓ | AP-1% ↑ | MME ↓ | AP-1% ↑ |
| $L_{CE}$ | 13.9 m [7.4 %] | 74.3 % | 88.0 m [11.7 %] | 88.6 % | 29.7 m [1.6 %] | 82.5 % |
| $L_{Dist}$ | 9.9 m [5.1 %] | 72.3 % | 119.5 m [15.9 %] | 85.5 % | 33.2 m [1.7 %] | 81.8 % |
| $L_{CE} + L_{Dist}$ | 9.1 m [4.9 %] | 74.3 % | 78.2 m [10.4 %] | 87.9 % | 29.3 m [1.5 %] | 83.5 % |

To validate the performance of our combined loss function, we conducted an ablation study. Specifically, we trained a model using only one component of the combined loss (cross-entropy $L_{CE}$ and distance loss $L_{Dist}$) and compare them to the results of the combined loss. The results are provided in Table 4.

For the FAU subset model, the distance loss improves the MME but not the AP-1%. Meanwhile, the cross-entropy is better for the AP-1% but not for the MME. The combination of both losses results in an improved MME while AP-1% remains the same. The results for the CReSIS subset are less clear. It is evident that the distance loss alone does not enhance the MME or AP-1%. However, the combined loss demonstrates the most optimal outcome in relation to the MME, while the AP-1% is only





slightly worse in comparison to CE alone. Similar to the CReSIS results, the distance loss alone does not improve the MME compared to the cross-entropy for the AWI subset. However, the combined loss again delivers the best results with a higher
AP-1% value.

## 5.4 Discussion and Outlook

One apparent influence on the quality of the ice bottom prediction is the primary thermal regime of the region. In general, the warmer the ice, the less reliable the prediction. The reason behind this is probably the influence of water on the signal, as well as the higher likelihood of a heavily crevassed surface. Temperate ice generally contains water, as most of the ice is
close to or at the pressure melting point. Water absorbs the recorded signal, leading to higher noise with increased depth and strong attenuation. Hence, the model's performance naturally decreases as the associated radargrams are more challenging to interpret. Polythermal glaciers, contrary to temperate glaciers, do not exhibit ice at the pressure melting point everywhere. Here, temperature is often induced into the ice due to strong frictional heat at regions of fast flow or close to the margins or the glaciers. Hence, the effects are not as detrimental as for entirely temperate glaciers.

Another striking observation is the difference between temperate and polythermal ice regarding the AP-1%. The AP-1% of temperate ice is significantly lower than for polythermal ice. However, this difference becomes a lot smaller when comparing the AP-5%. A possible explanation for this could lie in the meltwater at the base of the ice. Temperate ice more commonly collects meltwater at its base than polythermal ice. Since water absorbs the signal, the exact position (AP-1%) becomes difficult to identify. However, the general position (AP-5%) is still clear because the water is only at the base. Besides the thermal regime
and average depth, the presence of debris usually plays a significant role in radio-echo sounding. Interestingly, the quantitative results of JRI and Viedma indicate that the presence of debris did not play a major role in the model's performance compared to depth and thermal regime. However, we suspect that the numbers do not capture the effect of debris very well since the debris likely absorbed the signal entirely. Thus, the expert could not create ground truth labels for these parts, which makes the effect of debris on the model's performance not accurately measurable with numerical methods.

A notable feature of the CReSIS model predictions is ice boundary collapses. When in doubt, the model shows a bias toward predicting the ice bottom close to the or as the ice surface. One explanation could be that the CReSIS data is differentiated and thus represents only the change in amplitude. That makes it challenging to distinguish whether the peak of the ice surface and ice bottom overlap or the ice bottom is not visible. The problem gets further amplified by noise and artifacts, such as multiples. They can exhibit similar patterns as the ice bottom, making the model biased toward predicting the ice bottom as the ice surface
when in doubt.

Furthermore, we believe that the influence of the ice boundary collapse is also reflected by the quantitative analysis of the different CReSIS study sites. As West Antarctica generally contains thicker ice sheets than the Antarctic Peninsula, the difference between the ice boundaries significantly increases. Thus, a wrong prediction of the ice bottom as ice surface leads to a considerably higher MAE and MME for West Antarctica than the Antarctic Peninsula. However, the ice boundary collapse
is likely not the only reason for this effect as the AP-1% and AP-5% are also lower for West Antarctica than the Antarctica Peninsula. Hence, thicker ice sheets might be naturally more challenging.





Nonetheless, future research should address ice boundary collapses as they tremendously affect performance. Larger contexts or recurrent neural networks could help stabilize the predictions as they incorporate more information. Another interesting problem to explore is the performance drop from subset-specific models to the omni model. Our results indicate that the
domain shift between the subsets is too prominent for a simple omni model to catch up on all subset-characteristic features. Hence, models cannot utilize the full benefits of a larger dataset when they are recorded and processed differently. In particular, domain shift techniques could help with this challenge.

We believe that our framework is the first step towards a potential fully automated generation of ice thickness maps based on RES data. Our presented work could lay the foundation for validating survey data while in the field.

## 6  Conclusions

This paper presents the first benchmark framework for delineating the ice boundary in RES data. The included dataset "IceAnatomy" contains hundreds of kilometers of processed, labeled, and georeferenced RES data from three different sources (FAU, CReSIS, AWI). Since all sources employ a different radar system and processing methods, "IceAnatomy" offers a wide range of varying amplitude spectrums, depth resolutions, and width resolutions, making it applicable to a multitude of settings.
Furthermore, it also features different geographical factors, such as study sites and thermal regimes, allowing for in-depth analysis of the models and their behavior in different geographical scenarios.

To fairly compare different models in the future, we provide an official train and test split for each source of the dataset. This enables the development of not only an omni model trained on the entire dataset but also specialized subset-specific models on one of the three sources. We trained and evaluated a baseline model for each of these scenarios. In our experiments, the
subset-specific models provide the most promising results with MMEs of $2.1\,\mathrm{m}\,[1.2\%]$, $23.1\,\mathrm{m}\,[3.1\%]$, and $4.9\,\mathrm{m}\,[0.3\%]$ for the ice surface and $9.1\,\mathrm{m}\,[4.9\%]$, $78.2\,\mathrm{m}\,[10.4\%]$, and $29.3\,\mathrm{m}\,[1.5\%]$ for the ice bottom depending on the source.

With this benchmark framework, we hope to encourage other scientists to start working in this challenging and important research area. Deep learning models that extract the ice boundary can greatly speed up the processing of RES data. As a result, ice depth and, consequently, the subglacial topography can be determined more quickly after a field survey.

*Code and data availability.* The dataset is available at https://zenodo.org/records/14036897 (Dreier et al., 2024) and the implementation at https://doi.org/10.5281/zenodo.14038570 (Dreier, 2024).

## Appendix A:  Additional Hyperparameters

This section gives an overview of the hyperparameters in our employed U-Net from Chapter 4.2. The input dimension of our U-net is (1024,512,1) (H,W,1), which then gets scaled according to the depth level of the encoder or decoder. Inside
the network, we down- and upsample our feature map five times each while scaling the feature dimension according to the depth-level-dependant value of $[8, 16, 32, 64, 64, 128]$. To reduce the risk of overfitting, we also utilize dropout layers inside



the ResBlocks with a probability of $10\%$. For the loss function, we employed our proposed combined loss function. Since the numerical value of the distance loss is significantly higher than that of the classification loss, we had to weigh the individual components. In detail, we chose the weights $w_{s\_class} = 0.5$, $w_{b\_class} = 1.0$, $w_{s\_dist} = 0.05$, and $w_{b\_dist} = 0.1$ as they performed
the best in preliminary experiments.

*Author contributions.* **Marcel Dreier**: Conceptualization, Methodology, Software, Experiments, Project administration, Data Processing, Visualization, Writing - Original draft preparation. **Moritz Koch**: Data Collection, Data Processing, Data curating, Visualization, Writing - Original draft preparation. **Nora Gourmelon**: Conceptualization, Writing - Original draft preparation. **Norbert Blindow**: Data Collection, Data Processing, Writing - review & editing. **Daniel Steinhage**: Data Collection, Data Processing, Writing - review & editing. **Fei Wu**:
Writing - review & editing. **Thorsten Seehaus**: Supervision, Writing - review & editing. **Matthias Braun**: Supervision, Writing - review & editing. **Andreas Maier**: Supervision, Writing – review & editing. **Vincent Christlein**: Supervision, Validation, Writing - review & editing.

*Competing interests.* The authors declare none of the authors have any competing interests.

*Acknowledgements.* This research was funded by the Bayerisches Staatsministerium für Wissenschaft und Kunst within the Elite Network Bavaria with the Int. Doct. Program "Measuring and Modelling Mountain Glaciers in a Changing Climate" (IDP M3OCCA)), as well as the
German Research Foundation (DFG) project "Large-scale Automatic Calving Front Segmentation and Frontal Ablation Analysis of Arctic Glaciers using Synthetic-Aperture Radar Image Sequences (LASSI)" and the DFG project "Ice thickness, remote sensing and sensitivity experiments using ice-flow modelling for major outlet glaciers of the Southern Patagonian Icefield"(ITERATE) grant DFG BR 2105/29-1/FU 1032/12-1. The authors gratefully acknowledge the scientific support and HPC resources provided by the Erlangen National High Performance Computing Center (NHR@FAU) of the Friedrich-Alexander-Universität Erlangen-Nürnberg (FAU) under the NHR projects
b110dc and b194dc. NHR funding is provided by federal and Bavarian state authorities. NHR@FAU hardware is partially funded by the DFG – 440719683. We acknowledge the use of data and data products from CReSIS generated with support from the University of Kansas, NASA Operation IceBridge grant NNX16AH54G, NSF grants ACI-1443054, OPP-1739003, and IIS-1838230, Lilly Endowment Incorporated, and Indiana METACyt Initiative. Furthermore, we also thank the support of Alfred-Wegener-Institut Helmholtz-Zentrum für Polar- und Meeresforschung. (2016). Polar aircraft Polar5 and Polar6 operated by the Alfred Wegener Institute. Journal of large-scale research facilities
JLSRF, 2 (0), 87. doi: 10.17815/jlsrf-2-153. The authors would like to thank Aspen Technology, Inc. for providing licenses in the scope of the Aspen Technology, Inc. Academic Program.

In order to improve the legibility of the manuscript, the authors have used ChatGPT (https://chatgpt.com/) and DeepL Write (https://www.deepl.com/en/write) to look for alternative phrases. The output of this service was reviewed and edited by the authors as needed. The authors take full responsibility for the content of the presented manuscript.



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
