# Peer review of "Ice Anatomy: A Benchmark Dataset and Methodology for Automatic Ice Boundary Extraction from Radio-Echo Sounding Data"

_EGUsphere, 2024_

## Author Response (AR1)

FAU (Inf. 5) | Martensstr. 3 | D-91058 Erlangen

**Computer Science Department 5**

**Pattern Recognition Lab**

Marcel Dreier

Martensstrasse 3, D-91058 Erlangen Room 09.156 Phone +49 9131 85-28977

Marcel.dreier@fau.de https://lme.tf.fau.de/person/madreier/

Erlangen, April 30, 2025

**Dear Editor and Reviewers,**

We thank the reviewers and the editor for their helpful and thoughtful comments. Their time and effort spent reviewing our manuscript are greatly appreciated. Following their advice, we carefully adjusted the manuscript and added additional experiments to the appendix. In detail, we have:

- Expanded and clarified explanations that were previously difficult to grasp.
- Improved the coloring of Figs. 1 and 4 to 7.
- Added additional experiments to the appendix.
- Improved consistency in terminology and tone.
- Added additional citations to related work to give a broader context of previous work.
- Clarified claims about the novelty and impact of our work.
- Adjusted the technical explanations for a better reading experience.

With these changes, we believe we have significantly improved the quality of our manuscript, and we hope to meet the expectations of the reviewers.

We are looking forward to hearing from you.

Kind regards, Marcel Dreier (on behalf of all co-authors)

The following pages contain a list of editor's and reviewers' comments followed by our replies. The comments are sequentially numbered and associated with the corresponding reviewer. The replies may contain references to changes in the original manuscript, which are identified by a label consisting of a running number and followed by the label of the original comment in parentheses. The label links back to the original reviewer's comment within the manuscript. For instance, the reference **C2** (1.3), which is typeset in the manuscript margin, refers to the second change stemming from the third comment of the first reviewer. In addition, please note that the reference section is not displayed correctly in this change document, for length and reference verification as well as correct table and figure numbering please refer to the "manuscript version".

**Comments of the 1st Reviewer:**

- 1.1 The choice to standardize all radargrams to a height of 1024 pixels requires further justification, especially given the reduction
   in resolution this causes, which could potentially affect the precision of the derived ice boundaries. The manuscript should provide a more detailed rationale (possibly linked to computational efficiency) for this choice, considering the capabilities of the U-Net-like to process varying input shapes.
- Thank you for this comment. As the reviewer has correctly stated, the U-Net can technically handle any input resolution. However, having different input resolutions can lead to significant drops in accuracy. It also complicates and slows down processing as samples within a batch are usually assumed to be the same size. However, there are two common solutions to this problem. The first is patching, where the image is cut into patches of equal resolution. Since our networks need to view the full height to make a layer prediction, this method is not applicable to our case. The other option is resizing, where we resize the input to a standardized resolution. We chose this measure for our experiments. Resizing all radargrams to the maximum height would avoid inaccuracies due to downscaling but introduces its own issues: potential unnatural artifacts due to upscaling, a substantial increase in model depth needed to maintain an effective receptive field, and significantly higher computational costs. Considering all these reasons, we found a standardized height of 1024 pixels a good compromise. We have added a brief explanation of our reasoning in change 28.
- 1.2 The decision to not use an exclusive flight for the AWI testing subset due to significant variability among the radargrams is questionable. The inherent variability could, in fact, provide a rigorous real-world test scenario, which is crucial for assessing
  25 the robustness and adaptability of the model to new and varied environments (which should be the eventual goal of any benchmark dataset and the models developed based on them). A reevaluation of the testing subset choice is recommended to potentially enhance the findings.
- ▶ The reviewer raises an excellent point. In theory, we agree that having an exclusive test flight is preferable. However, all three AWI flights are from different campaigns with slightly different processing and different survey areas, introducing a domain shift. These changes, combined with the overall smaller size of the subset, make generalizations from one flight to another challenging without additional advanced techniques to mitigate the domain shift. Since these techniques would have been out of scope for this work and we still wanted to provide meaningful results for the AWI subset, we did not select an exclusive test flight for the AWI subset. Please note that generalization between different flights can still be tested with the other subsets.
- 1.3 The omni model shows reduced performance in the FAU and AWI domains, which the authors attribute to domain shifts. Consideration of alternative approaches such as weighting samples by domain frequency or uniformly sampling training examples across domains could potentially mitigate this issue. An exploration of these methods would be valuable for enhancing model generalization.
- ▶ This is an interesting point, which we investigated further with additional experiments in our appendix (see change 13). From the results, we can see a slight drop in performance for the CReSIS subset but substantial improvements for the AWI and FAU subsets compared to the original omni model. The model even outperforms its dataset-specific counterpart on the AWI subset. However, we still see a substantial performance gap in the FAU subset compared to the dataset-specific model. We reason that the FAU domain is naturally further away from the AWI and CReSIS domains, as it consists of undifferentiated radargrams. Thus, the uniform sampling strategy does help to mitigate the domain shift but does not fix it entirely when the processing of the radargrams differs significantly.

- **454** The proposed U-Net uses two heads to separately predict the ice surface and bottom. Why is it better than a straightforward approach with one head simultaneously doing both? Softmax can be applied later in the column-wise manner to extract the boundaries as well, so it should not be a limitation.
  - ▶ Thank you for your comment. The idea of having a separate head for every task is a common approach for multi-task models in deep learning across various categories.
- In our case, it allows every head to focus on a single layer instead of solving both simultaneously. As the reviewer correctly stated, the task can also be solved with a single head that projects onto multiple output channels depending on the specified prediction task. However, this also increases the size of the head proportionally to the output channels. In the case of the output head consisting of a single convolutional layer, the two approaches would be mathematically equivalent.
- 1.5 The authors write in Section 5.1: "Depending on the chosen method, the metrics used to assess the quality of the predictions differ," which is not really true, as zone predictions are easily convertible to boundaries and vice versa, so there is no problem to providing the whole set of metrics.
  - ▶ Thank you for your comment. It is true that we can easily calculate the zones from the boundary prediction. However, a metric like the IoU does not give precise intel about the layer extraction task. Thus, we did not include it in our evaluation as we mainly focus on layer extraction.
- However, going from zone predictions to layer predictions is a lot more difficult. In zone segmentation tasks, the model decides the zone class of each pixel separately. Hence, the resulting prediction does not need to have clearly identifiable layers. We give a small example in the figure below. In those cases, layer extraction and defining a corresponding error become ambiguous. Therefore, these tasks usually rely on segmentation metrics, as we cannot define layer metrics.

1.6 The manuscript claims that confusion matrix-based metrics would perform poorly if predictions are, e.g., consistently off by a
 65 pixel. However, this statement is misleading as these metrics are typically used for zone predictions, not boundary delineations.
 A correction or further explanation is needed to resolve this confusion.

- ▶ Thank you, we clarified this paragraph in the revised version of the paper (compare change 31).
- 1.7 In Appendix A, it is stated that the authors used dropout layers inside the ResBlocks. Was it a regular dropout? If not, it should be specified. If yes, I would suggest also trying something like spatial dropout, as many practitioners found it more helpful in convolutional networks.
  - ▶ Thank you for the advice. We have utilized a regular Dropout Layer in our implementation. We have not experimented with spatial Dropout before, but it will be an interesting avenue for further optimizations. We added spatial dropout to future work and clarified the use of our dropout layer (compare addition 48 and change 11).
- **1.8** Figures 6, 7, and similar graphics are challenging to interpret. I would suggest just plotting four curves—two groundtruths (surface and bottom) and two predictions on top (e.g. dashed).
  - ▶ Thank you for pointing this out. We adjusted the visualizations accordingly (compare Fig. 4, 5, 6, and 7).

**Comments of the 2nd Reviewer:**

- 2.1 The manuscript uses varying terminology, such as "the air-ice layer and ice-ground layer" and "ice bottom and ice surface layer." Maintaining consistency in terminology throughout the text would improve clarity and readability. ... Line 36: The phrase "radargram of the glacier" sounds somewhat awkward. Additionally, the manuscript does not always adhere consistently to standard glaciological terminology.
  - ▶ Thank you for this helpful suggestion. We unified our terminology (compare changes 3, 8, 4, 10, 19, and the update to the caption of Fig. 3).
- 2.2 The manuscript claims to present the first benchmark dataset for ice boundary extraction, yet related datasets such as CReSIS data have been widely used. The authors should explicitly contrast IceAnatomy with existing datasets and justify why this dataset is uniquely valuable beyond just being a "benchmark."
  - ▶ Thank you for raising this very important point. As correctly highlighted, IceAnatomy is not the first dataset in this field. However, it is the first benchmark dataset. Thus, it has to conform to higher standards, unlike previous datasets.
  - One such standard is reproducibility. While some works use the same dataset, they do not precisely define their training and testing subsets. Our evaluation on the CReSIS subset demonstrates that the composition of these splits can significantly influence model performance (see Table 1). For example, if the test set consisted only of radargrams from the Antarctic Peninsula, the model's performance would appear substantially better. This emphasizes the necessity of standardized training and testing splits, such as those provided in a benchmark dataset, to ensure a fair and meaningful comparison between multiple models.

Another important factor is generalization. IceAnatomy provides data spanning multiple campaigns, radar systems, institutions, and diverse glaciological environments. Thus, researchers can evaluate their model in a multitude of scenarios to achieve good generalization.

Lastly, institutions like CReSIS may provide a large public database for radargrams with an annotated ice bottom layer, but their label quality is not guaranteed to have human-level accuracy. CReSIS themselves state that they utilize snake trackers, leading edge detectors, interpolation, and peak detectors based on the judgment of the operator picking the data, making the quality difficult to judge (compare https://data.cresis.ku.edu/data/rds/rds\_readme.pdf). Although most labels are still correct, neural networks generally learn to imitate the annotation process, so models trained solely on automated labels may learn peak detection rather than more generalizable features. Thus, we need to have humanly annotated data. This might raise concerns about whether the CReSIS subset is an appropriate choice for a benchmark dataset. However, CReSIS has been the primary source of data in this field, and multiple works report human annotations for this specific dataset. Hence, we deem it appropriate to include it, even though some labels might be noisy.

We adjusted the manuscript to highlight the benefits of our new benchmark dataset (see addition 3).

- **2.3** The radargram visualizations are useful but could benefit from additional annotations. Additionally, the color scheme makes it difficult to distinguish certain features, and the way different annotations are represented could be improved for better clarity
- ▶ Thank you for pointing this out. We adjusted the visualizations to make them clearer and easier to grasp (compare the updated 110 Fig. 4, 5, 6, and 7).

- 2.4 The manuscript is highly technical and may be challenging for a glaciological audience. Since The Cryosphere primarily targets glaciologists, the extensive use of computer science jargon and technical terminology either requires more thorough explanations or suggests that a different journal may be a better fit. Ensuring the content is more accessible to the journal's primary readership should be a key consideration.
- Lines 240–245, 264–271, and other similar sections contain highly technical explanations. These should either be clarified and simplified for better accessibility to the journal's audience or, if the technical depth is essential, the choice of journal may need to be reconsidered.
- ▶ Thank you for raising this point. The editor initially asked us to add more explanations for the technical terms. As a result, some parts became a bit more technical than originally intended, as the explanations already require a certain level of expertise. We try to address this issue by simplifying our explanations and moving the more in-depth explanations in the Appendix. Please, see deletions 3 and 4, change 30 and the updated Appendix C and B.
- **2.5** The scientific motivation of the study could be further elaborated. This is one of the aspects that might suggest the paper, in its current form, would be better suited for a more technical journal.
- 125▶ Thank you for pointing that out. We have revised the manuscript to express our motivation and vision for the future more clearly (see change 1).
  - **2.6** The rationale for the baseline model choices (e.g., why U-Net with specific modifications) should be better justified. Why not test other architectures such as Transformers or hybrid CNN-RNN models?
- ▶ The U-Net is a widely adopted approach for tasks such as ice boundary extraction and comparable tasks. While more recent architectures like the transformer or recurrent neural networks may offer better performance, they also come with increased computational costs, larger models, and other practical limitations. This might hinder the deployment in the field and other time-critical scenarios. For that reason, we chose the U-Net as our baseline model. We added our reasoning in change 29.
- 2.7 The manuscript states that the dataset consists of manually labeled ice boundaries but does not provide sufficient details on the annotation process. What steps were taken to ensure label accuracy? Were multiple annotators involved? How was inter-annotator variability handled?
  - ▶ Thank you for your comment. We have clarified the picking process for the FAU subset in addition 5, for the AWI subset in addition 6, and added further details and references for the CReSIS picking procedure in change 25.
- **2.8** The inclusion of noisy annotations from CReSIS data is acknowledged, but how does this affect training and evaluation? Have any data cleaning techniques been applied?
- 140 Thank you for this comment. Several works have already pointed out that they consider the labels from the CReSIS subset noisy. Although CReSIS provided quality ratings for each label, we found several cases where the reasoning behind them was unclear to us. Below, we provide two example radargrams from the Abbott Ice Shelf. In the first radargram, several parts of the ice bottom are missing, although the original annotator has high confidence (green) in his labeled ice bottom. In the second example, the quality rating changes throughout the section between high confidence (green) and medium confidence (yellow)
- with no clear indication in the radargram. We included the CReSIS quality labels to stay consistent with the original. However, we do not think they necessarily reflect the mentioned noise. Thus, we could not evaluate how this noise affects the training or evaluation process.
- **2.9** The dataset includes different radar systems and processing methods, which may introduce domain shifts. Are these shifts quantified? How do they impact model performance?
- 150▶ Thank you for this question. As the reviewer has highlighted, the domain shift is a serious problem introduced by different radar systems, study sites, board electronics, and processing steps. As a result, it is nearly impossible to quantify the domain

shift in a precise mathematical manner. However, we have summarized the most critical processing steps in Section 3.2. From these steps, we can derive some qualitative characteristics, e.g., the AWI and CReSIS provide differentiated radargrams.

How does this impact the results? From our experiments with the Omni model, we can see that the data from different domains do not necessarily work well together. After conducting follow-up experiments in Appendix 13, we hypothesize that the issue might be connected to the differentiation of the radargram. However, we must also point out that the domain shift is very difficult to study as we cannot isolate this variable. We therefore deem it out of scope for this work and leave it to future studies to investigate this phenomenon in more detail.

- **2.10** The AP-5% metric relaxes the error bounds, but why were 1% and 5% chosen? Would alternative thresholds (e.g., 2% or 10%) provide additional insights?
  - ▶ Thank you for this comment. While we agree that offering additional intervals could potentially help us analyze the error better, adding too many would also make the evaluation and interpretation of the error very confusing. Hence, we chose the AP-1% and AP-5% as they represent a near-perfect and a good pick.
- **2.11** The "ice boundary collapse" issue observed in predictions is significant. Could this be mitigated with additional constraints in the loss function or post-processing techniques?
  - ▶ Thank you for your interest. We believe that modifying the loss function further is unlikely to significantly mitigate the boundary collapse, as our proposed distance-based loss already puts more emphasis on large outliers like the boundary collapse. However, additional post-processing steps could be an interesting approach for future work. The question here would be how

- to prevent the algorithm from overcorrecting traces when the boundary collapse takes up the majority of the transit. As this would require more experiments and testing, it is outside the scope of this work. However, we included post-processing as a potential method for future work to explore (compare addition 10).
- **2.12** The paper does not discuss the impact of hyperparameters in training. How sensitive is the model to learning rate, regularization, and architecture modifications?
- Thank you for this comment. We believe that the focus of this paper lies in the benchmark dataset rather than the benchmark model. Accordingly, our ablation study focuses on factors connected to the data such as the study site, the domain shift, and the thermal regime. For that reason, we chose the hyperparameters like the learning rate close to standard values for this field. However, we added a small ablation study in addition 14 to investigate two of the hyperparameters more closely. From the results, we can see that different hyperparameter setups favor different subsets of IceAnatomy. However, there seems to be no universal optimal setup.
- 2803 Some terms, such as "depth resolution," "relative error," and "wave velocity assumptions," need clearer definitions in the main text rather than just appearing in equations.
  - ... Line 315: The phrase 'pass through a pixel' is unclear. At times, the radargram are treated as an image, and at other times as a matrix. However, it is important to note that a wave does not pass through a pixel. .... Line 324: The argument presented is not compelling.
- 185 Thank you for pointing out the need for further clarification. We clarified the definitions of wave velocity and depth resolution in changes 33 and 32. We also rephrased our argumentation to make it clearer and more compelling. This also allowed us to further simplify the text and avoid the term relative error (compare change 35).
- **2.14** *Some parts of the manuscript have an informal tone.*
- ▶ Thank you for pointing this out. We adjusted several parts of the text to address this issue (see changes 1, 2, 5, 6, 7, 9, 4, 14, 190 18, 21, 20, 36, 37, 41, 42, and 46).
- **2.15** The manuscript overstates the novelty and impact of its contributions. It describes the framework as the "first step" toward automated ice thickness mapping, despite acknowledging decades of prior research. Similarly, the claim that this work has "invited other scientists to start working" in this area overlooks longstanding studies. These statements should be revised to more accurately reflect the field's history. ...
- Line 448: The statement "We believe that our framework is the first step towards a potential fully automated generation of ice thickness maps based on RES data" could be reworded for accuracy. As noted in the literature review, research in this area has been ongoing for nearly two decades. While this work is a valuable contribution, positioning it as the first step towards automation may not fully acknowledge prior advancements in the field.
- Line 462: The statement suggesting that this work has "invited other scientists to start working in this research area" may overstate its impact. Given the examples of previous studies provided by the authors, it would be more accurate to acknowledge the long-standing research efforts in this field while highlighting how this study builds upon them.
  - ▶ Thank you for pointing this out. Our intention was not to discredit previous research in this area. We adjusted the wording of the sentences (compare changes 49 and 50)
- **2.16** Line 98: Jebeli et al. 2023 have performed a very similar aim to this work in their study.
- 205 Line 88: Moqadam et al. 2024 (DOI: 10.22541/essoar.172987463.39597493/v1) also have done the tracking of internal layers.
  - Line 98 105: The manuscript would benefit from citing additional relevant work to provide a more complete context for readers. For instance, Moqadam and Eisen (https://doi.org/10.5194/egusphere-2024-1674) offers a broad review of prior research on ice boundary extraction, making it a fitting reference at the end of the literature review.
- 210 Line 102: Where the use of CNN for autoamtic tracing of internal layers is mentioned, Jebeli et al. 2023

(DOI: 10.13140/RG.2.2.23219.20007), Moqadam et al. 2024 (DOI: 10.22541/essoar.172987463.39597493/v1) directly addresses the application of deep learning to this task and would be valuable citations in the section discussing recent advancements in this area.

Including these references, along with other relevant studies would help situate the manuscript within the broader body of existing research and provide readers with a more comprehensive view of the field.

- ▶ Thank you for pointing this out. We included the missing references to previous work in Section 2.
- **2.17** Line 64: The statement, "however, a large portion of ...," should be supported with evidence. Importantly, the critique of automatically labeled bedrock seems contradictory, as the study itself aims to achieve this. Clarifying this point would strengthen the argument.
- 220 Thank you for highlighting this, as this is an important point. We do not intend to criticize datasets for utilizing automatic approaches to label their ice boundaries. However, we do criticize the use of such datasets in the context of training and evaluating deep learning approaches. Deep learning models essentially learn to imitate the labeling process. If all the labels are based on a peak detection algorithm, then our model learns peak detection. This defeats the purpose of employing deep learning models, which is to achieve more accurate predictions. Thus, we should avoid datasets where the labeling process was
- significantly automized or lacked the necessary transparency. We rephrased the text and added citations as an example. Please view changes 2 and 11.
- **2.18** *Line 124: not clear what the authors want to say.*
  - ▶ Thank you for this comment. We have rephrased the text to make it clearer in change 13.
- **2.19** Line 134 137: More references are needed to support the claims
- 230 Thank you for pointing this out. We have added further references in this section to support our claim.
- **2.20** Line 141: "Hence, ... " it is not clear or accurate argument for the clearer signal of the thinner ice. The aim of the sentence is evident but the sentence should be reformulated.
  - ▶ Thank you for bringing this to our attention. We have slightly adjusted the text in change 16.
- **2.21** *Line 148: the sentence seems to be incomplete.*
- 235 Thank you for bringing this to our attention. We have adjusted the manuscript accordingly in change 17.
- **2.22** *Line 181: this process needs to be elaborated.*
  - ▶ Thank you for this comment. We have expanded on our explanation in change 22.
- **2.23** Line 226: "the" should be removed.
  - Line 251: hyphen needed between differently and sized.
- 240 Thank you for pointing this out. We have removed the extra 'the' and added the missing hyphen.
- **2.24** *Line 381: The sentence needs to be rewritten for clarity.*
  - ▶ Thank you for bringing this to our attention. We rephrased the specified sentence in change 38.
- **2.25** Line 311: the authors mention that resizing changes the MAE so they introduce MME. It is not clear why they keep the MAE in the paper, if MME is a more suitable metric.
- 245▶ Thank you for this comment. We added an explanation in the text (compare addition 8).
- **2.26** *Lines* 392-399: *These sentences need to be rewritten for clarity and flow.*
  - ▶ Thank you for bringing this to our attention. We have adjusted the text to improve clarity and flow (compare change 39).

- **2.27** *Line 401: Please provide a more detailed explanation of the ablation study.*
  - ▶ Thank you for this comment. We expanded on our explanation of the ablation study in change 40.
- Line 401: Please provide a more detailed explanation of the ablation study.
- 2.28 Line 414: the explanation of temperate ice this can appear much earlier in the manuscript
  - ▶ Thank you for pointing this out. We added an earlier explanation in change 12.
- **2.29** *line 418: the sentence should be rewritten.*
  - ▶ Thank you for pointing this out. We changed the sentence in change 43.
- **2530** Line 421: It is obvious that the differences decrease when AP-5% is considered, and there is nothing surprising about this result. Please rewrite this statement or clarify the reasoning behind the argument.
  - ▶ Thank you for mentioning this. You are correct that a decrease in error difference has to be expected, but our point was to highlight the strong decrease. We rephrased our point to make it clearer. Please view change 44.
- **2.31** *Line 430: the sentence does not read well.*
- 260▶ Thank you for this comment. We rephrased the sentence (compare change 45).
- **2.32** Line 437: Please provide further explanation. What exactly do you mean, and why is this the case?
  - ▶ Thank you for this comment. We adjusted the sentence to make it clearer (compare change 47). Naturally, the ice surface and ice bottom are generally further apart in thicker ice.
- **2.33** Line 441: The authors mention that thicker ice is more challenging, but shouldn't it actually be easier, as less collapse would occur in thicker ice compared to thinner ice?
  - While we agree that the presence of shallow ice in the dataset first introduces the ice boundary collapse, once the model picks up on this characteristic, it also affects the predictions of thicker ice. The problem of the ice boundary collapse ultimately stems from the nature of differentiated radargrams, where we only visualize the change in amplitude. Here, peaks indicate a change in amplitude and are usually connected to the ice surface or ice bottom. When there is no or only very shallow ice, the peaks for the ice surface and bottom start overlapping, and it is often unclear to the model whether the two peaks overlap or whether it cannot find the peak for the ice bottom. Thus, it develops a bias to predict the ice bottom as the ice surface whenever it cannot find the second peak corresponding to the ice bottom.
- Why does this affect thicker ice more than shallow ice? In thicker ice, the returned signal of the radar system is generally weaker than in shallow ice due to attenuation and more potential sources of interference. Thus, the depicted peak for the ice bottom becomes less clear. As a result, the model now faces considerable difficulty in determining whether the weak peak corresponds to the true ice bottom or if the ice surface and bottom overlap. We also added this explanation to the manuscript in addition 9.

**Ice Anatomy: A Benchmark Dataset and Methodology for Automatic Ice Boundary Extraction from Radio-Echo Sounding Data**

Marcel Dreier1, Moritz Koch2, Nora Gourmelon1, Norbert Blindow2, Daniel Steinhage3, Fei Wu1, Thorsten Seehaus2, Matthias Braun2, Andreas Maier1, and Vincent Christlein1

**Correspondence:** Marcel Dreier (marcel.dreier@fau.de)

280

285

290

300

**Abstract.** The measurement of ice thickness is of great importance for the accurate estimation of glacier volume and the delineation of their bedrock topography. In particular, this is a crucial factor in forecasting the future evolution of glaciers in the context of a changing climate. In order to derive the ice thickness, the travel time of electromagnetic waves in radargrams acquired by radio-echo sounding (RES) systems is analyzed. This can only be achieved by identifying the ice surface and underlying ice bottom in corresponding radargrams. Manually identifying these two reflection horizons in RES data is a laborious and timeconsuming process. Consequently, scientists are attempting to automate this task through the use of techniques such as deep learning. Such automation can significantly reduce the time between a field campaign and the calculation of the glacier's ice thickness distribution. In this paper, we present the first benchmark dataset for delineating the ice surface and bottom boundaries in RES data, to facilitate straightforwardstandardized comparisons of deep learning models in the future. The "IceAnatomy"

C1 (2.14) dataset comprises radargrams and the corresponding manual picks, amounting to a total of over 45,000 km of observations. The RES data originates from three sources: FAU, CReSIS, and AWI. The dataset comprises different RES systems as well as different pre-processing methods. In addition, the data was acquired over a large range of geographical and glaciological settings, featuring different thermal regimes present in Antarctica and the Southern Patagonian Icefield. This diversity ensures that the models' behaviors can be analyzed in different scenarios. We define a standardized train-test split for each source in the dataset. This allows us to introduce not only a baseline model trained on the entire training set (the "omni" model), but also three source-specific baseline models. The source-specific models are trained exclusively on the subset of the training data acquired by the specified source. The baseline models provide an initial benchmark against which subsequent models can be compared. The source-specific models demonstrate more accurate results than the omni model. For the FAU, CReSIS, and AWI test sets, the source-specific models achieve Mean Meter Errors of 2.1 m, 23.1 m, and 4.9 m for the ice surface and 9.1 m, 78.2 m, and 29.3 m for the ice bottom. In relation to the mean measured ice thickness of the test set, these errors equate to 1.2%, 3.1%, and 0.3% for the ice surface and 4.9%, 10.4%, and 1.5% for the ice bottom. The dataset and implementation are available at https://zenodo.org/records/14036897 (Dreier et al., 2024) and https://doi.org/10.5281/zenodo.14038570 (Dreier, 2024).

<sup>1Department of Computer Science, Friedrich-Alexander-Universität Erlangen-Nürnberg, Erlangen, Germany.

<sup>2Institut für Geographie, Friedrich-Alexander-Universität Erlangen-Nürnberg, Erlangen, Germany.

<sup>3Alfred Wegener Institute for Polar and Marine Research, Bremerhaven, Germany.

**1 Introduction**

305

310

320

325

330

Glaciers and ice shelves are key indicators of global climate (Haeberli et al., 2007; IPCC, 2013). Knowing their volume and ice thickness distribution is crucial for assessing future cryospheric contributions to sea level rise. Moreover, data on the ice volume of glaciers and ice sheets is necessary for understanding their response to climate change. Ice thickness measurements enable the subsequent prediction of the rate and timing of glacier retreat or disappearance using different types of models. That way, a glacier's contribution to regional hydrological cycles and subsequent influence on local to regional scales with associated socioeconomic impacts can be assessed This enables the assessment of a glacier's contribution to regional hydrological cycles and its subsequent influence on local to regional scales with associated socioeconomic impacts. (Werder et al., 2020; Ayala ГC2 (2.14) et al., 2020; Farinotti et al., 2017). Several techniques to determine ice thickness exist, including seismic, gravitational, and magnetic methods, as well as radio-echo sounding (RES) (Bogorodsky et al., 2012; Kohler et al., 1997). While satellite gravimetry allows for a resolution in the range of kilometers, its spatial resolution does not allow for the interpretation of detailed subglacial features (Willen et al., 2024). Seismic measurements offer a high resolution, but widespread use in the Antarctic region is limited by high exploration costs or logistical unfeasibility (An et al., 2023). For this reason, RES is preferred over other methods when an accurate assessment of a subglacial topography is of interest. After pre-processing the RES data, a cross-section, a so-called radargram of the glacier, becomes visible. After pre-processing the RES data, we obtain an image commonly referred to as a radargram. It depicts the cross-section of the glacier along the flight path. Experts can then interpret the RES data by delineating the reflections of surfaces or internal glacial structures. Delineating the ice boundary, defined by the air-ice layer and ice-ground layerice surface (air to ice transition) and the ice bottom (ice to ground/water transition), is necessary to obtain the glacier's thickness at each point in the radargram. However, it is a laborious time-intensive [ task, especially with large datasets (Sime et al., 2011). Several automated and semi-automated approaches to delineate the layers have been developed (Fahnestock et al., 2001; Gifford et al., 2010; Freeman et al., 2010; Rahnemoonfar et al., 2017a, b; Kamangir et al., 2018; Rahnemoonfar et al., 2019; Cai et al., 2020, 2022; Liu-Schiaffini et al., 2022b; Moqadam and Eisen, 2024; Moqadam et al., 2024; Jebeli et al., 2023b). However, these approaches are not comparable as they have been evaluated on different datasets or a different train-test split of the same dataset. In this paper, we present a publicly available, ready-to-use standardized benchmark dataset for ice thickness extraction. It is the first of its kind to be directly conjured designed for deep learning approaches, with a pre-defined train-test split, human-annotated labels, and different recording systems. It comprises radargrams from Antarctica and Patagonia with polythermal, cold-based, or temperate thermal regimes. The dataset is intended for supervised training and evaluation of deep learning models. Therefore, the dataset includes depth labels for both the ice bottom and ice surface layer. Together with the dataset, we present a baseline model that delineates the ice boundary in a given radargram. The model is based on the U-Net architecture (Ronneberger et al., 2015) and serves as a reference and a starting point for future improvements. We envision further development of this method in two main directions. First, once trained, our algorithm can be executed on virtually any modern laptop in the field. Combined with a pre-processing chain tailored to our approach, this allows for near real-time analysis of acquired data on-site. Since flight hours are costly and often limited by weather conditions, optimizing their use is crucial. If data can be processed in the field—e.g., between two flights—flight

plans could be dynamically adjusted to focus on areas of high interest within the same campaign. Second, and perhaps more importantly, the presented method can be further developed to handle more specialized tasks, such as delineating intraglacial water channel systems or identifying water tables within existing datasets. This would represent a step toward a comprehensive, quantitative, and standardized approach for interpreting radargrams, ultimately leading to fully automated products that could significantly benefit the cryospheric research community. In particular, the automated mapping of internal reflection layers remains a critical knowledge gap – one that deep learning is well – positioned to address (Moqadam and Eisen, 2024).

In summary, our contributions are as follows:

- 1. A novel benchmark dataset IceAnatomy for deep learning-based extraction of ice boundary from RES data is created.
- 2. A baseline deep learning model for the automatic delineation of ice bottom and ice layer the ice bottom and the ice surface is proposed.
- 345 3. A thorough evaluation of individual models and an omni-model is conducted on the dataset.

The work is structured as follows: Section 2 provides an overview of datasets and algorithms previously used for automatic ice boundary extraction. Subsequently, Sect. 3 gives insight into the recording and processing of the dataset as well as relevant geographical and glaciological factors of the study sites. The baseline models are introduced in Sect. 4. An extensive evaluation of the baseline models and the benchmark dataset is presented in Sect. 5. Lastly, we summarize our research and draw conclusions in Sect. 6.

**2 Related Works**

350

Over the past decades, RES has been widely used in glaciology. A multitude of publications cover the extraction of ice boundary layers from RES data. In this section, we highlight related RES datasets and layer extraction approaches.

**2.1 Datasets**

RES data on glaciers and ice sheets is abundantly available. Numerous RES datasets on glaciers and ice sheets are publicly available. However, a large portion of the associated bedrock labelsice bottom labels are inaccurate, generated automatically, or unavailable, unavailable, lack the necessary transparency in their creation, or do not have associated radargrams (Young et al., 2021; Blankenship et al., 2018; CReSIS; Dong et al., 2022; Corr, 2020; Corr et al., 2020). This makes them unsuitable for training or evaluating deep learning approaches, as they require human-annotated data. Hence, we focus our comparison on datasets that have been used to extract the ice boundary in previously published work and for which both radargrams and human-annotated labels are publicly available. These constraints significantly limit the number of related datasets.

[revised manuscript text omitted]

. This saves computing power while keeping all the essential information. To restore the full flight | C21 (2.14) 485 traces in the FAU dataset from their subsampled parts, we reassembled the radargrams according to their trace numbers. Any conflicting depths for the ice surface and bottom in overlapping parts were smoothened with Gaussian importance weighting. Furthermore, we filled gaps of eleven pixels or less in the ice surface and bottom via bicubic interpolation. Furthermore, in rare cases, the initially labeled ice surface and bottom had small gaps. To avoid such inconsistencies, we filled gaps of eleven pixels or less in the ice surface and bottom via bicubic interpolation, using the two nearest manually labeled points as a reference. 490 The initial layer labels were annotated by a single interpreter to ensure consistency throughout the dataset. Surface reflections were generally straightforward to identify; however, in heavily crevassed areas, we increased the resolution to delineate the airice interface as accurately as possible across these features. Bedrock picks were conducted using the same approach. In regions with ambiguous reflections, ReflexW software enabled zooming into specific subsets of the radargrams, thereby enhancing the clarity of features of interest. Additionally, several intersecting profile lines provided cross-points for internal validation. 495 These intersections were annotated independently by the same interpreter and subsequently compared. All cross-profile values fell within the expected margin of error, even in areas with steep slopes or greater depths (i.e., deviations < 10 %). At Glacier Perito Moreno, two control points from previous studies were available for comparison (Sugiyama et al., 2011; Stuefer et al., 2007). The first, along the 'Buscaini' profile, corresponds to a seismic survey conducted in 1996, which reported a maximum ice thickness of 720 m. The second, located nearer to the glacier terminus, corresponds to a borehole drilled in 2010, revealing

**3.2.2 CReSIS Data**

500

505

The CReSIS data was recorded during the 2009 campaign of Operation Ice Bridge in Antarctica, which comprised 21 missions. Three were sea-ice surveys and thus are not included in the CReSIS dataset. The remaining 18 missions can be split into two groups: six missions focusing on the Antarctic Peninsula (PEN1, PEN2, PEN3, PEN4, PEN5, and LVISPEN) and 12 missions exploring West Antarctica (PIG1, PIG2, PIG3, PIG4, LVISPIG, LVIS86, GETZ1, ABBOTT1, TSK1, TSK2, TSK3, and TSK4) (Allen et al., 2012b). All 18 missions employed the Multichannel Coherent Radar Depth Sounder (MCoRDS) flown on a McDonnell Douglas DC-8-72. It has a center frequency of 195 MHz and an eight-channel-chirp signal to accurately assess the groundice (Rodriguez-Morales et al., 2014; Shi et al., 2010b).

an ice thickness of  $515 \pm 5$  m. Both control points were in close agreement with our ice thickness estimates.

C23 (2.1)

To process the recorded data, the standard CReSIS L1B CSARP-mvdr (minimum variance distortionless response) processing steps were applied. These include pulse compression via a Tukey and Hanning Window, beam-forming, motion compensation, synthetic aperture radar processing in combination with f-k migration, channel combination, and waveform combination (CReSIS, 2024b). After the processing, the radargrams had a depth resolution of  $105 \, \mathrm{ns}$  pixel $^{-1}$  and a width resolution of  $12 - 30 \, \mathrm{m}$  pixel $^{-1}$  depending on the mission.

[revised manuscript text omitted]

As metrics for our benchmark framework, we have chosen the MAE, two relaxed Average Precision (AP) metrics, and introduce the Mean Meter Error (MME). The MAE is calculated as the column-wise difference in pixels between the ground truth depth of a layer and the predicted depth. Resizing the radargram will change the value of this metric. Therefore, we

**Figure 4.** A visual representation of the four metrics used in this work. The left side of the figure depicts the MAE and MME respectively as the difference between prediction and ground truth (GT). Meanwhile, the right side of the figure features the AP-1% and AP-5% respectively as an interval around the ground truth. Note that the ground truth and the predictions are technically float numbers. However, we thickened the ground truth by 20 pixels to improve visibility.

also introduce the MME, which approximates the real-world error. We calculate the MME by multiplying the MAE with the product of the wave velocity in the medium and the depth resolution of the radargram. The speed of the wave in the medium is assumed to be constant with the speed of light ( $c_{air} = 0.299792458 \,\mathrm{m \, ns^{-1}}$ ) for the air layer and  $c_{ice} = 0.168 \,\mathrm{m \, ns^{-1}}$  for the iee-layer. The wave velocity describes the speed of the electromagnetic wave of the radar through a medium. We assume it to be constant with the speed of light ( $c_{air} = 0.299792458 \,\mathrm{m \, ns^{-1}}$ ) in air and with  $c_{ice} = 0.168 \,\mathrm{m \, ns^{-1}}$  in ice (Johari and Charette, 1975). The depth resolution is the time it takes for the wave to pass through a pixelthrough the physical equivalent of a pixel in the radargram. Since the depth resolution is indirectly proportional to the y-dimension of the radargram, the MME stays consistent across different heights. Table 1 records the different depth resolutions for radargrams in the IceAnatomy dataset in their original height and equation 6 and 7 summarize the formula for the MME. Note that the MME is still highly dependent on the original depth resolution of the radargram. The MME will be naturally higher for a radargram where every pixel constitutes a 40 m change in height rather than a 4 m change, as even small mistakes lead to a drastic increase. Thus, we also record the MAE as it is more consistent over radargrams of the same image height but with different study sites and radar systems.

$$MME_{s}(\hat{y}_{s}, y_{s}) = \frac{c_{air}}{2|\hat{Y}_{s}|} \sum_{\hat{y}_{s} \in \hat{Y}_{s}} Depth-Reso^{*} \cdot MAE_{s}(\hat{y}_{s}, y_{s})$$

$$(6)$$

$$MME_{b}(\hat{y}_{b}, y_{b}) = \frac{c_{ice}}{2|\hat{Y}_{b}|} \sum_{\hat{y}_{b} \in \hat{Y}_{b}}^{g_{s} \in Y_{b}} Depth-Reso^{*} \cdot MAE_{b}(\hat{y}_{b}, y_{b})$$

$$(7)$$

(\* Depth-Resolution after resizing the radargram.)

MME and MAE both describe the distance between two lines. A disadvantage is that they are not robust to outliers. As an outlier robust alternative, we also use a relaxed average precision (AP). To normalizestandardize the relaxation, we count everything below a 1 or 5% error of the total height in pixels of the radargram as a hit (AP-1% and AP-5%). Choosing a relative error bound instead of fixing it to an absolute pixel value prevents the metric from changing when the radargram is resized. For our chosen height of 1024 pixels, this would mean the AP-1% allows for an error of 10.24 pixels, and the AP-5% allows for an error of 51.2 pixels. Tying the average precision to the height of the radargram prevents the metric from drastically changing if future studies resize the radargrams differently. In addition, relaxing the metric alleviates the problem of uncertainties in the labels. Figure 4 shows a visualization of the employed metrics.

**5.2 Experimental Protocol**

[revised manuscript text omitted]

combinations. Still, dataset and model-specific discrepancies exist.

**5.3.1** Ice Surface Predictions**

The predictions for the ice surfaces are nearly perfect for all subsets and all models. The three subset models even achieve 100% accuracy for the AP-5%. Hence, the remaining discrepancies are likely significantly influenced by measurement inaccuracies, noise, and general model variance. Therefore, we will only consider the task of ice bottom delineation to assess model performance.

**5.3.2** Ice Bottom Predictions**

[revised manuscript text omitted]

**755 5.3.5 Loss Function**

760

To validate the performance of our combined loss function, we conducted an ablation study. Specifically, we trained a model using only one component of the combined loss (cross-entropy  $L_{CE}$  and distance loss  $L_{Dist}$ ) and compare them to the results of the combined loss. The results are provided in Table 4. To assess the performance of our combined loss function, we conducted a small ablation study. Specifically, we evaluated two additional experiments in which we replaced the combined loss with each of its individual components: In the first setup, we trained the model with the cross-entropy loss, and in the second setup, we trained it only with the distance loss. We compare the results of these two configurations with the combined loss in Table 4.

-C40 (2.27)

For the FAU subset model, the distance loss improves the MME but not the AP-1%. MeanwhileIn contrast, the cross— C41 (2.14) entropy is better for the AP-1% but not for the MME. The combination of both losses results in an improved MME while

AP-1% remains the same. The results for the CReSIS subset are less clear. It is evident that the distance loss alone does not enhance the MME or AP-1%. However, the combined loss demonstrates the most optimal outcome in relation to the MME, while the AP-1% is only slightly worse in comparison to CE alone. Similar to the CReSIS results, the distance loss alone does not improve the MME compared to the cross-entropy for the AWI subset. However, the combined loss again deliversprovides — C42 (2.14) the best results with a higher AP-1% value.

**5.4 Discussion and Outlook**

770

780

785

790

795

One apparent influence on the quality of the ice bottom prediction is the primary thermal regime of the region. In general, the warmer the ice, the less reliable the prediction. The reason behind this is probably the influence of water on the signal, as well as the higher likelihood of a heavily crevassed surface. Temperate ice generally contains water, as most of the ice is close to or at the pressure melting point. Water absorbs the recorded signal, leading to higher noise with increased depth and strong attenuation. Hence, the model's performance naturally decreases as the associated radargrams are more challenging to interpret. Polythermal glaciers, contrary to temperate glaciers, do not exhibit ice at the pressure melting point everywhere. Here, temperature is often induced into the ice due to strong frictional heat at regions of fast flow or close to the margins or the glaciers. Instead, elevated temperatures are usually confined to zones of fast flow driven by frictional heating or to marginal areas of the glacier. Hence, the effects are not as detrimental as for entirely temperate glaciers.

-C43 (2.29)

Another striking observation is the difference between temperate and polythermal ice regarding the AP-1%. The AP-1% of temperate ice is significantly lower than for polythermal ice. However, this difference becomes a lot smaller when comparing the AP-5%. Another interesting observation is the difference between temperate and polythermal ice regarding the AP-1% and AP-5%. The AP-1% of temperate ice is significantly lower than for polythermal ice. However, the AP-5% is relatively similar for both types of ice. While it is natural for the difference to decrease at higher error intervals, the change in this case is still very drastic. To put this into perspective, the Viedma and James Ross Island were 16.4 percentage points apart on the AP-1%, but on the AP-5%, only 0.1 percentage points. A possible explanation for this could lie in the meltwater at the base of the ice. Temperate ice more commonly collects meltwater at its base than polythermal ice. Since water absorbs the signal, the exact position (AP-1%) becomes difficult to identify. However, the general position (AP-5%) is still clear because the water is only at the base. Besides the thermal regime and average depth, the presence of debris usually plays a significant role in radio-echo sounding. Interestingly, the quantitative results of JRI and Viedma indicate that the presence of debris did not play a major role in the model's performance compared to depth and thermal regime. However, we suspect that the numbers do not capture the effect of debris very well since the debris likely absorbed the signal entirely. Thus, the expert could not create ground truth labels for these parts, which makes the effect of debris on the model's performance not accurately measurable with numerical methods.

A notable feature of the CReSIS model predictions is ice boundary collapses. One of the more prominent and recurring phenomena in the CReSIS model's predictions is the collapse of ice boundaries. When in doubtIn ambiguous cases, the model

-C46 (2.14)

shows a bias toward predicting the ice bottom close to the or as the ice surface. One explanation could be that the CReSIS data is differentiated and thus represents only the change in amplitude. That makes it challenging to distinguish whether the peak of the ice surface and ice bottom overlap or the ice bottom is not visible. The problem gets further amplified by noise and artifacts, such as multiples. They can exhibit similar patterns as the ice bottom, making the model biased toward predicting the ice bottom as the ice surface when in doubt. Thicker ice sheets are particularly affected by ice boundary collapse, as the radar signal returned from deeper ice is typically weaker than in shallower regions. This is due to increased attenuation and a higher chance of signal interference. Consequently, the peaks representing the ice bottom in the radargram become less distinct. As a result, the model encounters significant challenges in discerning whether a weak peak indicates the actual ice bottom or if it is just noise, and the signals from the surface and bottom overlap.

Furthermore, we believe that the influence of the ice boundary collapse is also reflected by the quantitative analysis of the different CReSIS study sites. As West Antarctica generally contains thicker ice sheets than the Antarctic Peninsula, the difference between the ice boundaries significantly increases. As West Antarctica generally contains thicker ice sheets than the Antarctic Peninsula, the average distance between the ice boundaries significantly increases. Thus, a wrong prediction of the ice bottom as ice surface leads to a considerably higher MAE and MME for West Antarctica than the Antarctic Peninsula. However, the ice boundary collapse is likely not the only reason for this effect as the AP-1% and AP-5% are also lower for West Antarctica than the Antarctica Peninsula. Hence, thicker ice sheets might be naturally more challenging.

—A9 (2.33)

-C48 (1.7)

Nonetheless, future research should address ice boundary collapses as they tremendously affect performance. Larger contexts, additional post-processing steps, or recurrent neural networks could help stabilize the predictions as they incorporate  $\Gamma^{A10}$  (2.11) more information. Another interesting problem to explore is the performance drop from subset-specific models to the omni model. Our results indicate that the domain shift between the subsets is too prominent for a simple omni model to catch up on all subset-characteristic features. Hence, models cannot utilize the full benefits of a larger dataset when they are recorded and processed differently. In particular, domain shift techniques could help with this challenge. In particular, domain shift techniques niques could help with this challenge, but also more advanced regularization techniques, e. g., spatial dropout, could prevent the model from focusing too much on a single domain (Tompson et al., 2015). In appendix 13, we show that a uniform sampling strategy can also help mitigate the domain shift.

We believe that our framework is the first step towards a potential fully automated generation of ice thickness maps based on RES data. Our presented work could lay the foundation for validating survey data while in the field. We believe that our framework is a significant step towards a potential fully automated generation of ice thickness maps based on RES data and that our work represents an important advancement toward validating survey data in the field.

**6 Conclusions**

800

805

810

815

820

825

This paper presents the first benchmark framework for delineating the ice boundary in RES data. The included dataset "IceAnatomy" contains hundreds of kilometers of processed, labeled, and georeferenced RES data from three different sources (FAU, CReSIS, AWI). Since all sources employ a different radar system and processing methods, "IceAnatomy" offers a wide range of varying amplitude spectrums, depth resolutions, and width resolutions, making it applicable to a multitude of settings. Furthermore, it also features different geographical factors, such as study sites and thermal regimes, allowing for in-depth analysis of the models and their behavior in different geographical scenarios.

To fairly compare different models in the future, we provide an official train and test split for each source of the dataset. This enables the development of not only an omni model trained on the entire dataset but also specialized subset-specific models on one of the three sources. We trained and evaluated a baseline model for each of these scenarios. In our experiments, the subset-specific models provide the most promising results with MMEs of 2.1 m [1.2%], 23.1 m [3.1%], and 4.9 m [0.3%] for the ice surface and 9.1 m [4.9%], 78.2 m [10.4%], and 29.3 m [1.5%] for the ice bottom depending on the source.

With this benchmark framework, we hope to encourage other scientists to start working in this challenging and important research area. Previous work has already demonstrated the effectiveness of automatic approaches for ice boundary extraction but lacked a common method for accurately comparing models. With this benchmark framework, we hope to address this issue by unifying and standardizing both training and evaluation schemes. We hope that this benchmark dataset will encourage more scientists to engage in this challenging and important research area. Deep learning models that extract the ice boundary can greatly speed up the processing of RES data. As a result, ice depththe ice thickness and, consequently, the subglacial consequently topography can be determined more quickly after a field survey.

845 *Code and data availability.* The dataset is available at https://zenodo.org/records/14036897 (Dreier et al., 2024) and the implementation at https://doi.org/10.5281/zenodo.14038570 (Dreier, 2024).

**Appendix A: Additional Hyperparameters**

835

840

850

855

This section gives an overview of the hyperparameters in our employed U-Net from Chapter 4.2. The input dimension of our U-net is (1024,512,1) (H,W,1), which then gets scaled according to the depth level of the encoder or decoder. Inside the network, we down- and upsample our feature map five times each while scaling the feature dimension according to the depth-level-dependant value of [8,16,32,64,64,128]. To reduce the risk of overfitting, we also utilize dropout layers inside the ResBlocks with a probability of 10%. For the loss function, we employed our proposed combined loss function. Since the numerical value of the distance loss is significantly higher than that of the classification loss, we had to weigh the individual components. In detail, we chose the weights  $w_{\text{s\_class}} = 0.5$ ,  $w_{\text{b\_class}} = 1.0$ ,  $w_{\text{s\_dist}} = 0.05$ , and  $w_{\text{b\_dist}} = 0.1$  as they performed the best in preliminary experiments.

**Appendix B: ResBlock Design**

To provide a better understanding of the network architecture, this section examines one of its core components: the ResBlock from (Esser et al., 2020). Its structure, shown in Fig. B1, comprises several components. First, it starts with a group normalization layer (Wu and He, 2018) that normalizes the data in groups of channels to increase stability during training. Next, a swish

activation (Ramachandran et al., 2017) function adds nonlinearity to the ResBlock so the network can learn more complex patterns. The activation is followed by a two-dimensional convolution layer that processes and combines the visual features by applying convolutional operations. This is followed by another group normalization and swish activation function before a regular dropout layer (Hinton et al., 2012) is applied. The dropout layer randomly withholds information during training to improve generalization and prevent the model from overfitting – a process in which the model develops a strong bias towards the training data. After the dropout layer, another two-dimensional convolutional layer is applied. Finally, a residual connection (He et al., 2016), a shortcut from the start of the ResBlock to the end through a convolution layer, is added to the output of this sequence of layers to improve the gradient flow in the network.

-A11 (1.7, 2.4)

**Figure B1.** Structure of the residual block employed in our deep learning model. The arrangement is based on the design of Esser et al. (2020)

**Appendix C: Loss Function Details**

The formulas of the classification and distance-based loss are as follows:

870
$$L_{\text{CE}} = -\sum_{c \in C} x_c (1 - \epsilon_c) \log(p(x_c)) + \frac{\epsilon_c (1 - x_c) \log(p(x_c))}{|C|}$$
(C1)

$$L_{\text{class}} = \frac{w_{\text{s\_class}}}{|\hat{Y}_{\text{s}}|} \sum_{\hat{y}_{\text{s}} \in \hat{Y}_{\text{s}}} L_{\text{CE}}(\hat{y}_{\text{s}}) + \frac{w_{\text{b\_class}}}{|\hat{Y}_{\text{b}}|} \sum_{\hat{y}_{\text{b}} \in \hat{Y}_{\text{b}}} L_{\text{CE}}(\hat{y}_{\text{b}})$$
(C2)

$$L_{\text{dist}} = \frac{w_{\text{s\_dist}}}{|\hat{Y}_{\text{s}}|} \sum_{\hat{y}_{\text{s}} \in \hat{Y}_{\text{s}}} \langle d(y_{\text{s}}), \sigma(\hat{y}_{\text{s}}) \rangle + \frac{w_{\text{b\_dist}}}{|\hat{Y}_{\text{b}}|} \sum_{\hat{y}_{\text{b}} \in \hat{Y}_{\text{b}}} \langle d(y_{\text{b}}), \sigma(\hat{y}_{\text{b}}) \rangle$$
(C3)

 $w_{\text{s\_class}}, w_{\text{b\_class}}, w_{\text{s\_dist}}$ , and  $w_{\text{b\_dist}}$  are the respective weights for a weighted combination of the single loss parts,  $\epsilon_c$  is the smoothing factor, C specifies the column,  $\langle \rangle$  is the dot product,  $\sigma$  is the softmax function that converts the model's outputs into

**Appendix D: Additional Experiments**

875

880

890

Since the three subsets of IceAnatomy differ in size, we also investigate whether a uniform sampling strategy, where samples are drawn equally from each subset, could help the Omni-Model achieve the performance of the domain-specific models on the AWI and FAU subsets. From our results in Table D1, we can see that a uniform sampling strategy does lead to improvement for the AWI and FAU subsets. In the case of the AWI subset, the omni model even outperforms the domain-specific model. However, in the case of the FAU subset, we are still below the domain-specific model. We reason that the domain of the AWI and CReSIS subsets are significantly closer than the FAU subset as these two subsets contain differentiated radargrams. We, therefore, believe that domain shift remains an important area for future research. In addition to the uniform sampling, we also  $\Gamma^{A13}$  (13.29)

**Table D1.** Overview of the performance of our Omni Model with uniform sampling. We distinguish the layer prediction into two classes: the ice surface (S) and the ice bottom (B). We compare the model's performance on the MME, MAE, AP-1%, and AP-5% as defined in Section 5.1. To contextualize the MME, we annotate the relative error to the mean measured ice thickness of the specified test set study site behind the MME. We conducted the evaluation on the test set and averaged the results over five runs to minimize statistical errors.

|        |       | Omni Model                 |       |                |                |  |  |  |
|--------|-------|----------------------------|-------|----------------|----------------|--|--|--|
|        | Layer | $\mathbf{MME}\downarrow$   | MAE ↓ | AP-1% ↑ | AP-5% ↑ |  |  |  |
| FAU    | S     | 2.0 m [1.1 %]              | 1.9   | 99.3%          | 100.0%         |  |  |  |
|        | В     | $14.0\mathrm{m}~[7.6\%]$   | 19.0  | 74.1%          | 94.1%          |  |  |  |
| CReSIS | S     | 23.1 m [3.1 %]             | 2.5   | 97.2%          | 100.0%         |  |  |  |
|        | В     | $75.0\mathrm{m}\;[10.0\%]$ | 14.6  | 87.7%          | 93.9%          |  |  |  |
| AWI    | S     | 3.8 m [0.2 %]              | 0.5   | 99.7,%         | 100.0%         |  |  |  |
|        | В     | $23.9\mathrm{m}\;[1.3\%]$  | 6.0   | 86.1%          | 98.3%          |  |  |  |

investigated how different hyperparameter setups regarding learning rate and regularization would affect the benchmark model.

From the results in Table D2 and D3, we can see that different hyperparameter setups favor different subsets of IceAnatomy.

However, there seems to be no universal optimal setup.

Author contributions. Marcel Dreier: Conceptualization, Methodology, Software, Experiments, Project administration, Data Processing, Visualization, Writing - Original draft preparation. Moritz Koch: Data Collection, Data Processing, Data curating, Visualization, Writing - Original draft preparation. Nora Gourmelon: Conceptualization, Writing - Original draft preparation. Norbert Blindow: Data Collection, Data Processing, Writing - review & editing. Daniel Steinhage: Data Collection, Data Processing, Writing - review & editing. Fei Wu:

**Table D2.** Overview of the performance of our Omni Model with different learning rates and uniform sampling. We distinguish the layer prediction into two classes: the ice surface (S) and the ice bottom (B). We compare the model's performance on the MME and AP-1% as defined in Section 5.1. To contextualize the MME, we annotate the relative error to the mean measured ice thickness of the specified test set study site behind the MME. We conducted the evaluation on the test set and averaged the results over three runs to minimize statistical errors. Note that for lr = 0.005 we averaged over five runs, as we had those values from previous experiments.

|        |       | lr = 0.0001                        |                | lr = 0.0005                |                | lr = 0.001                |                |
|--------|-------|------------------------------------|----------------|----------------------------|----------------|---------------------------|----------------|
|        | Layer | $\mathbf{MME}\downarrow$           | AP-1% ↑ | MME ↓                      | AP-1% ↑ | MME ↓                     | AP-1% ↑ |
| FAU    | S     | 2.1 m [1.1 %]                      | 99.0%          | 2.0 m [1.1 %]              | 99.3%          | 2.1 m [1.1 %]             | 99.1%          |
|        | В     | $14.1\mathrm{m}\;[7.6\%]$          | 73.9%          | $14.0\mathrm{m}\;[7.6\%]$  | 74.1%          | $14.3\mathrm{m}\;[7.7\%]$ | 74.1%          |
| CReSIS | S     | 26.2 m [3.5 %]                     | 96.7%          | 23.1 m [3.1 %]             | 97.2%          | 21.9 m [2.9 %]            | 97.6%          |
|        | В     | $105.4\mathrm{m}\;[14.0\%]$        | 87.2%          | $75.0\mathrm{m}\;[10.0\%]$ | 87.7%          | $94.9\mathrm{m}[12.6\%]$  | 87.9%          |
| AWI    | S     | 4.4 m [0.2 %]                      | 99.5%          | 3.8 m [0.2 %]              | 99.7, %        | 3.7 m [0.2 %]             | 99.6%          |
|        | В     | $26.9\mathrm{m}\left[1.4\%\right]$ | 86.2%          | $23.9\mathrm{m}\;[1.3\%]$  | 86.1%          | $21.5\mathrm{m}\;[1.1\%]$ | 87.8%          |

**Table D3.** Overview of the performance of our Omni Model with different learning rates and uniform sampling. We distinguish the layer prediction into two classes: the ice surface (S) and the ice bottom (B). We compare the model's performance on the MME and AP-1% as defined in Section 5.1. To contextualize the MME, we annotate the relative error to the mean measured ice thickness of the specified test set study site behind the MME. We conducted the evaluation on the test set and averaged the results over three runs to minimize statistical errors. Note that for dropout = 0.1 we averaged over five runs, as we had those values from previous experiments.

[revised manuscript text omitted]

---

## Referee Report (RR1)

**Review of "Ice Anatomy: A Benchmark Dataset and Methodology for Automatic Ice Boundary Extraction from Radio-Echo Sounding Data"**

This manuscript presents a benchmark dataset, "IceAnatomy," designed to support and standardize the development and evaluation of deep learning models for extracting ice surface and bottom boundaries from radio-echo sounding (RES) radargrams. The dataset includes over 45,000 km of RES observations from multiple institutions and systems across diverse glaciological settings, along with baseline models and standardized train-test splits. Overall, the work addresses a pressing need in the cryosphere and remote sensing communities for reproducible, large-scale datasets that can accelerate progress in automated ice thickness estimation.

I commend the authors for their thorough and careful revisions, which have significantly improved the clarity, completeness, and overall quality of the manuscript since the previous review round.

Below, I provide detailed comments regarding the strengths and areas where the manuscript could be improved.

- In the sentence "It is the first..." line 45, the term "human-annotated labels" could be clarified further. Does this refer to fully manual annotations or semi-automated labels subsequently verified or correctd by humans? Given the importance of label quality in training and benchmarking deep learning models, this distinction is relevant for understanding the dataset's reliability.
- Line 51: please rewrite the sentence
- lines 52-75: The authors suggest that near real-time identification of the ice bottom boundary during RES data acquisition could allow for dynamic adjustments of flight plans to focus on areas of high interest. While this is an interesting idea, I wonder how often knowledge of the **ice bottom alone**, without broader context (e.g., basal conditions, surface conditions, prior survey goals), would justify altering flight plans during a campaign. Some clarification or examples from field experience would strengthen this claim and help the reader better understand its practical relevance.
- Line 57: "This would represent a step toward a comprehensive, quantitative, and standardized approach for interpreting radargrams, ultimately leading to fully automated products that could significantly benefit the cryospheric research community." please clarify that "interpreting radargrams" is **only in terms of ice surface and ice bottom boundaries**.
- Line 75 77: The statement regarding the limitations of existing ice bottom labels is broad and lacks sufficient specificity. To strengthen this important critique, the authors should clearly separate and elaborate on each claimed issue, such as inaccuracies, automatic generation methods, data unavailability, lack of transparency, and missing radargrams, and provide concrete examples or citations of datasets where these problems have been documented. Without this clarification, the claim risks appearing vague and unsubstantiated, which weakens the justification for the need and novelty of the IceAnatomy dataset. More precise and evidence-backed discussion is necessary to convincingly demonstrate the dataset's advantages over existing resources.

- Line 77: In support of the statement regarding the limitations of existing ice bottom labels (e.g., inaccuracy, lack of transparency, or missing radargrams), the authors cite several references. However, Dong et al. 2022 use **synthetic radargrams**, which may not be directly relevant to a critique of real RES datasets or their associated manual/automatic annotations. I recommend revisiting this citation and ensuring that each reference clearly supports the specific issue being discussed. This would improve the precision of the argument and strengthn the manuscript's positioning.
- Line 102 108: In the list of references for works that track internal ice and snow layers, the citation *Moqadam and Eisen (2024)* is included alongside algorithm-focused studies. However, this is a **review article** rather than a method paper, so it may be better to distinguish it from the rest. Consider adding a sentence such as "*For an overview of methods used in this domain, see Moqadam and Eisen (2024)*" instead. This would clarify the nature of the citation and improve the precision of the literature summary.
- Line 141: The statement "As the glaciers are temperate, i.e., most of the ice is close to or at the pressure melting point, they contain a relatively high proportion of water" would benefit from a supporting reference. Please consider citing glaciological studies or datasets that characterize the thermal regime and water content of these specific glaciers to substantiate this claim.
- Line 143: The authors state that the glacier characteristics "pose a significant challenge to machine learning systems.". That is very good intuition, I appreciate that. This is an important and plausible point, but it would be helpful to clarify whether this is based on **prior research**, **quantitative comparisons** in the current study, or anecdotal experience. If other studies have demonstrated lower model performance on temperate glaciers or radargrams from deep/steep troughs, please cite them. Otherwise, consider softening the language or providing some evidence from the dataset or baseline results presenteed in this paper.
- Line 154: The reference to *Rignot et al.* (2011) for ice velocity maps is valid, but more recent and higher-resolution velocity datasets are now available. I recommend updating or complementing this citation with a more recent source to ensure the comparison reflects the current state of ice velocity mapping.

**• Line 203 – 213:**

- The authors provide a commendably detailed description of the annotation process, including the use of a single interpreter for consistency, cross-profile validation, and comparison with control points. This level of detail strengthens confidence in the dataset quality, good job.
- Since the authors mention using *ReflexW* software for zooming and clarifying radargram features, it would be helpful to include a formal citation or reference for this commercial software to guide readers unfamiliar with it.
- The description suggests that the labeling involved some degree of software-assisted (semi-automatic) annotation rather than purely manual picking. For clarity, please

specify whether the labels were created fully manually, semi-automatically with manual corrections, or a combination thereof. This clarification is important for users evaluating the dataset and its annotations.

- Line 268 270: The description of the U-Net–based model for ice boundary extraction is clear and well supported by relevant citations. However, I suggest including an additional recent relevant work in this context: *Moqadam et al. (2024)*, which presents a closely related approach with U-Net for ice boundary extraction using deep learning. Also, as the cited version of this work is a preprint, please update the citation to the published version to ensure readers have access to the finalized paper.
- Line 341: "depth resolution is the time it takes for the wave to pass through the physical equivalent of a pixel in the radargram," is not scientifically accurate. Depth resolution refers to the minimum vertical distance between two subsurface reflectors that can be distinguished as separate features in the radargram. It is a spatial (distance) parameter, not a temporal one, and depends on the radar wave velocity and the system's temporal (time) resolution. I recommend revising this sentence for clarity and accuracy.
- Line 487: Tone of self-evaluation. The sentence claiming the work is "a significant step" and "an important advancement" could benefit from more objective framing or clearer support from the results. Consider revising this statement to maintain a more neutral tone in line with scientific conventions. While the impact of the work is indeed notable, I suggest moderating this language unless further evidence is provided to substantiate such claims in comparison to existing datasets or methods.

---

## Author Response (AR2)

FAU (Inf. 5) | Martensstr. 3 | D-91058 Erlangen

**Computer Science Department 5**

**Pattern Recognition Lab**

Marcel Dreier

Martensstrasse 3, D-91058 Erlangen Room 09.156

Phone +49 9131 85-28977

Marcel.dreier@fau.de https://lme.tf.fau.de/person/madreier/

Erlangen, August 14, 2025

Dear Editor and Reviewers,

We thank the reviewers and the editor for their thoughtful and constructive feedback that helped us improve our manuscript. In response to their suggestions, we have carefully revised the paper and made some minor adjustments. Specifically, we have:

- Added a conversion table from pixels to meters in the Appendix.
- Revised parts that were still unclear.
- Updated our references.

With these changes, we believe we have further improved the quality of our manuscript, and we hope to meet the expectations of the reviewers and the editor. We are looking forward to hearing from you.

Kind regards,

Marcel Dreier (on behalf of all co-authors)

The following pages contain a list of editor's and reviewers' comments followed by our replies. The comments are sequentially numbered and associated with the corresponding reviewer. The replies may contain references to changes in the original manuscript, which are identified by a label consisting of a running number and followed by the label of the original comment in parentheses. The label links back to the original reviewer's comment within the manuscript. For instance, the reference C2 (1.3), which is typeset in the manuscript margin, refers to the second change stemming from the third comment of the first reviewer. In addition, please note that the reference section is not displayed correctly in this change document, for length and reference verification as well as correct table and figure numbering please refer to the "manuscript version".

**Comments of the 1st Reviewer:**

- **1.1** The authors addressed all comments in full, and the paper is ready to be published as is.
- 10 We are glad to hear that the reviewer considers our manuscript to meet the journal's standards for publication, and we sincerely appreciate their valuable feedback to help improve our manuscript.
- 1.2 still, however, have one small technical recommendation. While the authors discussed the (computational) reasons for resizing all radargrams to 1024 px height, a small sentence regarding what limitations it has in terms of the limited resolution of the final predictions would be appreciated. Also, to aid readers in interpreting the quantitative results, it would be helpful to state
   15 the conversion "I px = x m" for each dataset (e.g. in the caption or as an extra column/row in the metric tables).
  - ▶ Thank you for this comment. We agree that providing a simple ratio like '1m = X px' can aid intuitive understanding. Thus, we added an explanatory section in the Appendix (compare addition 9). In the same section, we also explain how the limited resolution affects the final prediction.

**Comments of the 2nd Reviewer:**

- Review of "Ice Anatomy: A Benchmark Dataset and Methodology for Automatic Ice Boundary Extraction from Radio-Echo Sounding Data" This manuscript presents a benchmark dataset, "IceAnatomy," designed to support and standardize the development and evaluation of deep learning models for extracting ice surface and bottom boundaries from radio-echo sounding (RES) radargrams. The dataset includes over 45,000 km of RES observations from multiple institutions and systems across diverse glaciological settings, along with baseline models and standardized train-test splits. Overall, the work addresses a pressing need in the cryosphere and remote sensing communities for reproducible, large-scale datasets that can accelerate progress in automated ice thickness estimation. I commend the authors for their thorough and careful revisions, which have significantly improved the clarity, completeness, and overall quality of the manuscript since the previous review round.
  - ▶ We thank the reviewer for their praise and their acknowledgment of the importance of our work. Furthermore, we would like to express our gratitude to the reviewers for providing invaluable feedback, allowing us to improve our manuscript further.
- 302 In the sentence "It is the first..." line 45, the term "human-annotated labels" could be clarified further. Does this refer to fully manual annotations or semi-automated labels subsequently verified or corrected by humans? Given the importance of label quality in training and benchmarking deep learning models, this distinction is relevant for understanding the dataset's reliability.
- ▶ The reviewer raises an important point. The ice bottom was, to the best of our knowledge, fully manually annotated. Please note that we do not know the person who annotated the CReSIS data; thus, we rely on the accounts from Lee et al. (2014) and Crandall et al. (2012). The ice surface is usually straightforward to spot, which makes semi-automated trackers with subsequent human correction and verification sufficient. We know that the annotations for the ice surface of the AWI radargrams were processed in that manner. However, the ice surface of the FAU was fully manually annotated, and according to Lee et al. (2014) and Crandall et al. (2012), the ice surface of the CReSIS subset was also fully manually annotated. We adjusted the text to reflect the annotation style better (refer to change 1).
- **2.3** *Line 51: please rewrite the sentence*
  - ▶ Thank you for bringing this up. We adjusted the sentence (refer to change 2).

- 2.4 lines 52-75: The authors suggest that near real-time identification of the ice bottom boundary during RES data acquisition could allow for dynamic adjustments of flight plans to focus on areas of high interest. While this is an interesting idea, I
  45 wonder how often knowledge of the ice bottom alone, without broader context (e.g., basal conditions, surface conditions, prior survey goals), would justify altering flight plans during a campaign. Some clarification or examples from field experience would strengthen this claim and help the reader better understand its practical relevance
- The reviewer raises an excellent point. Of course, the idea can vary depending on the campaign objectives. However, during our past fieldwork in Patagonia, we were often interested in specific areas with features beneath the ice, which we could only assume beforehand, often due to changes in surface morphology or similar indicators. While flight planning depends on the research objectives, we have, on several occasions, identified features of interest or their full extent only after processing the data. These could include, for example, a bedrock ridge beneath a glacier whose extent is greater than initially assumed, or ice inflow from large tributary glaciers into the main valley, which alters the bedrock topography at points of conflux more than expected. On the other hand, if conditions such as heavy snowfall in or near the accumulation zone attenuate the signal and make it impossible to identify bedrock features, subsequent flights can be adjusted. In such cases, we might shift focus away from that area and instead survey another region more specifically or with denser coverage. We included examples in the text (compare addition 1).
- 2.5 Line 57: "This would represent a step toward a comprehensive, quantitative, and standardized approach for interpreting radar-grams, ultimately leading to fully automated products that could significantly benefit the cryospheric research community."60 please clarify that "interpreting radargrams" is only in terms of ice surface and ice bottom boundaries.
  - ▶ Thank you for pointing this out. We clarified the sentence in change 3.
- 2.6 Line 75 77: The statement regarding the limitations of existing ice bottom labels is broad and lacks sufficient specificity. To strengthen this important critique, the authors should clearly separate and elaborate on each claimed issue, such as inaccuracies, automatic generation methods, data unavailability, lack of transparency, and missing radargrams, and provide concrete examples or citations of datasets where these problems have been documented. Without this clarification, the claim risks appearing vague and unsubstantiated, which weakens the justification for the need and novelty of the IceAnatomy dataset. More precise and evidence-backed discussion is necessary to convincingly demonstrate the dataset's advantages over existing resources.
  - ▶ Thank you for this comment. We revised the text to provide more details about each point. Please refer to change 4.
- Z07 Line 77: In support of the statement regarding the limitations of existing ice bottom labels (e.g., inaccuracy, lack of transparency, or missing radargrams), the authors cite several references. However, Dong et al. 2022 use synthetic radargrams, which may not be directly relevant to a critique of real RES datasets or their associated manual/automatic annotations. I recommend revisiting this citation and ensuring that each reference clearly supports the specific issue being discussed. This would improve the precision of the argument and strengthn the manuscript's positioning.
- 75 Thank you for this comment. It is true that Dong. et al. employ synthetic data to train their models. However, they also validate their model on field data obtained from the Kunlun station from the 29th Chinese Antarctic Scientific Expedition. Unfortunately, this data is not publicly accessible to our knowledge, which makes it, in our opinion, a fitting reference.
- 2.8 Line 102 108: In the list of references for works that track internal ice and snow layers, the citation Moqadam and Eisen (2024) is included alongside algorithm-focused studies. However, this is a review article rather than a method paper, so it may be better to distinguish it from the rest. Consider adding a sentence such as "For an overview of methods used in this domain, see Moqadam and Eisen (2024)" instead. This would clarify the nature of the citation and improve the precision of the literature summary.
  - ▶ Thank you for bringing this to our attention. Moqadam and Eisen (2024) is indeed a review article rather than a separate method paper. We adjusted the text accordingly (refer to addition 2).

- 859 Line 141: The statement "As the glaciers are temperate, i.e., most of the ice is close to or at the pressure melting point, they contain a relatively high proportion of water" would benefit from a supporting reference. Please consider citing glaciological studies or datasets that characterize the thermal regime and water content of these specific glaciers to substantiate this claim.
  - ▶ Thank you for this comment. We included additional citations (compare addition 3).
- 2.10 Line 143: The authors state that the glacier characteristics "pose a significant challenge to machine learning systems."
  90 That is very good intuition, I appreciate that. This is an important and plausible point, but it would be helpful to clarify whether this is based on prior research, quantitative comparisons in the current study, or anecdotal experience. If other studies have demonstrated lower model performance on temperate glaciers or radargrams from deep/steep troughs, please cite them. Otherwise, consider softening the language or providing some evidence from the dataset or baseline results presenteed in this paper.
- 95 Thank you for this suggestion. While we are not aware of similar studies that describe these difficulties, we found that, especially in our datasets of temperate outlet glaciers, the radargrams contained more noise. This may be attributed to signal attenuation due to high water content and signal scattering caused by the sometimes extreme surface roughness, in the form of crevasses. We clarified the text in addition 4.
- **2.11** Line 154: The reference to Rignot et al. (2011) for ice velocity maps is valid, but more recent and higher-resolution velocity datasets are now available. I recommend updating or complementing this citation with a more recent source to ensure the comparison reflects the current state of ice velocity mapping.
  - ▶ Thank you for this comment. We included an additional reference (compare addition 5).
- 2.12 Line 203 213: The authors provide a commendably detailed description of the annotation process, including the use of a single interpreter for consistency, cross-profile validation, and comparison with control points. This level of detail strengthens
   confidence in the dataset quality, good job.
  - ▶ Thank you very much for the commendation of our work. We are glad to hear that the transparency of our annotation process strengthens confidence in the dataset quality.
- 2.13 Since the authors mention using ReflexW software for zooming and clarifying radargram features, it would be helpful to include a formal citation or reference for this commercial software to guide readers unfamiliar with it. The description suggests that the labeling involved some degree of software-assisted (semi-automatic) annotation rather than purely manual picking. For clarity, please specify whether the labels were created fully manually, semi-automatically with manual corrections, or a combination thereof. This clarification is important for users evaluating the dataset and its annotations.
  - ▶ Thank you for this comment. We added a reference for the ReflexW software and clarified that all the annotations in the FAU dataset were done fully manually, except for the gap interpolation at the end (compare addition 6 and 7).
- 21.54 Line 268 270: The description of the U-Net-based model for ice boundary extraction is clear and well supported by relevant citations. However, I suggest including an additional recent relevant work in this context: Moqadam et al. (2024), which presents a closely related approach with U-Net for ice boundary extraction using deep learning. Also, as the cited version of this work is a preprint, please update the citation to the published version to ensure readers have access to the finalized paper.
- ► Thank you for pointing this out. At the time of writing, only the preprint was available. We have now updated the corresponding references and included Moqadam et al. in the list of references for the U-Net.
- 2.15 Line 341: "depth resolution is the time it takes for the wave to pass through the physical equivalent of a pixel in the radargram," is not scientifically accurate. Depth resolution refers to the minimum vertical distance between two subsurface reflectors that can be distinguished as separate features in the radargram. It is a spatial (distance) parameter, not a temporal one, and depends on the radar wave velocity and the system's temporal (time) resolution. I recommend revising this sentence for clarity and accuracy.

- ▶ Thank you for this comment. After careful review, we realised the term depth resolution does not exactly describe what we wanted to express. Instead, we replaced the term depth resolution with the more fitting term two-way travel time resolution (compare changes 5, 6, 7, 8, 9, 10, 11, 12, 13, and 14.)
- 2.16 Line 487: Tone of self-evaluation. The sentence claiming the work is "a significant step" and "an important advancement"130 could benefit from more objective framing or clearer support from the results. Consider revising this statement to maintain a more neutral tone in line with scientific conventions. While the impact of the work is indeed notable, I suggest moderating this language unless further evidence is provided to substantiate such claims in comparison to existing datasets or methods.
  - ▶ Thank you for this suggestion. We moderated our language (please refer to changes 1 and 2).

**Ice Anatomy: A Benchmark Dataset and Methodology for Automatic Ice Boundary Extraction from Radio-Echo Sounding Data**

Marcel Dreier1, Moritz Koch2, Nora Gourmelon1, Norbert Blindow2, Daniel Steinhage3, Fei Wu1, Thorsten Seehaus2, Matthias Braun2, Andreas Maier1, and Vincent Christlein1

**Correspondence:** Marcel Dreier (marcel.dreier@fau.de)

**Abstract.** The measurement of ice thickness is of great importance for the accurate estimation of glacier volume and the delineation of their bedrock topography. In particular, this is a crucial factor in forecasting the future evolution of glaciers in the context of a changing climate. In order to derive the ice thickness, the travel time of electromagnetic waves in radargrams acquired by radio-echo sounding (RES) systems is analyzed. This can only be achieved by identifying the ice surface and underlying ice bottom in corresponding radargrams. Manually identifying these two reflection horizons in RES data is a laborious and time-consuming process. Consequently, scientists are attempting to automate this task through the use of techniques such as deep learning. Such automation can significantly reduce the time between a field campaign and the calculation of the glacier's ice thickness distribution. In this paper, we present the first benchmark dataset for delineating the ice surface and bottom boundaries in RES data, to facilitate standardized comparisons of deep learning models in the future. The "IceAnatomy" dataset comprises radargrams and the corresponding manual picks, amounting to a total of over 45,000 km of observations. The RES data originates from three sources: FAU, CReSIS, and AWI. The dataset comprises different RES systems as well as different pre-processing methods. In addition, the data was acquired over a large range of geographical and glaciological settings, featuring different thermal regimes present in Antarctica and the Southern Patagonian Icefield. This diversity ensures that the models' behaviors can be analyzed in different scenarios. We define a standardized train-test split for each source in the dataset. This allows us to introduce not only a baseline model trained on the entire training set (the "omni" model), but also three source-specific baseline models. The source-specific models are trained exclusively on the subset of the training data acquired by the specified source. The baseline models provide an initial benchmark against which subsequent models can be compared. The source-specific models demonstrate more accurate results than the omni model. For the FAU, CReSIS, and AWI test sets, the source-specific models achieve Mean Meter Errors of 2.1 m, 23.1 m, and 4.9 m for the ice surface and 9.1 m, 78.2 m, and 29.3 m for the ice bottom. In relation to the mean measured ice thickness of the test set, these errors equate to 1.2%, 3.1%, and 0.3% for the ice surface and 4.9%, 10.4%, and 1.5% for the ice bottom. The dataset and implementation are available at https://zenodo.org/records/14036897 (Dreier et al., 2024) and https://doi.org/10.5281/zenodo.14038570 (Dreier, 2024).

<sup>1Department of Computer Science, Friedrich-Alexander-Universität Erlangen-Nürnberg, Erlangen, Germany.

<sup>2Institut für Geographie, Friedrich-Alexander-Universität Erlangen-Nürnberg, Erlangen, Germany.

<sup>3Alfred Wegener Institute for Polar and Marine Research, Bremerhaven, Germany.

**1 Introduction**

160

175

180

185

190

Glaciers and ice shelves are key indicators of global climate (Haeberli et al., 2007; IPCC, 2013). Knowing their volume and ice thickness distribution is crucial for assessing future cryospheric contributions to sea level rise. Moreover, data on the ice volume of glaciers and ice sheets is necessary for understanding their response to climate change. Ice thickness measurements enable the subsequent prediction of the rate and timing of glacier retreat or disappearance using different types of models. This enables the assessment of a glacier's contribution to regional hydrological cycles and its subsequent influence on local to regional scales with associated socioeconomic impacts. (Werder et al., 2020; Ayala et al., 2020; Farinotti et al., 2017). Several techniques to determine ice thickness exist, including seismic, gravitational, and magnetic methods, as well as radioecho sounding (RES) (Bogorodsky et al., 2012; Kohler et al., 1997). While satellite gravimetry allows for a resolution in the range of kilometers, its spatial resolution does not allow for the interpretation of detailed subglacial features (Willen et al., 2024). Seismic measurements offer a high resolution, but widespread use in the Antarctic region is limited by high exploration costs or logistical unfeasibility (An et al., 2023). For this reason, RES is preferred over other methods when an accurate assessment of a subglacial topography is of interest. After pre-processing the RES data, we obtain an image commonly referred to as a radargram. It depicts the cross-section of the glacier along the flight path. Experts can then interpret the RES data by delineating the reflections of surfaces or internal glacial structures. Delineating the ice boundary, defined by the ice surface (air to ice transition) and the ice bottom (ice to ground/water transition), is necessary to obtain the glacier's thickness at each point in the radargram. However, it is a time-intensive task, especially with large datasets (Sime et al., 2011). Several automated and semi-automated approaches to delineate the layers have been developed (Fahnestock et al., 2001; Gifford et al., 2010: Freeman et al., 2010: Rahnemoonfar et al., 2017a, b; Kamangir et al., 2018: Rahnemoonfar et al., 2019: Cai et al., 2020, 2022; Liu-Schiaffini et al., 2022b; Mogadam and Eisen, 2025; Mogadam et al., 2025; Jebeli et al., 2023b). However, these approaches are not comparable as they have been evaluated on different datasets or a different train-test split of the same dataset. In this paper, we present a publicly available, standardized benchmark dataset for ice thickness extraction. It is the first of its kind to be directly designed for deep learning approaches, with a pre-defined train-test split, human-annotated labelsfully human annotated ice bottom labels, and different recording systems. It comprises radargrams from Antarctica and Patagonia with polythermal, cold-based, or temperate thermal regimes. The dataset is intended for supervised training and evaluation of deep learning models. Therefore, the dataset includes depth labels for both the ice bottom and ice surface layer. Together with the dataset, we present a baseline model that delineates the ice boundary in a given radargram. The model is based on the U-Net architecture (Ronneberger et al., 2015) and serves as a reference and a starting point for future improvements. We envision further development of this method in two main directions. In the following, we highlight two potential areas where our algorithm could be further extended to impact real-world scenarios in the future. First, once trained, our algorithm can be executed on virtually any modern laptop in the field. Combined with a pre-processing chain tailored to our approach, this allows for near real-time analysis of acquired data on-site. Since flight hours are costly and often limited by weather conditions, optimizing their use is crucial. If data can be processed in the field—e.g., between two flights—flight plans could be dynamically adjusted to focus on areas of high interest within the same campaign, e.g. a bedrock ridge beneath a glacier whose

extent is greater than initially assumed or ice inflow from large tributary glaciers into the main valley, which alters the bedrock topography at points of conflux more than expected. Second, and perhaps more importantly, the presented method can be further developed to handle more specialized tasks, such as delineating intraglacial water channel systems or identifying water tables within existing datasets. This would represent a step toward a comprehensive, quantitative, and standardized approach for interpreting radargrams interpreting the ice surface and ice bottom in radagrams, ultimately leading to fully automated products that could significantly benefit the cryospheric research community. In particular, the automated mapping of internal reflection layers remains a critical knowledge gap – one that deep learning is well – positioned to address (Moqadam and Eisen, 2025).

In summary, our contributions are as follows:

- 1. A novel benchmark dataset IceAnatomy for deep learning-based extraction of ice boundary from RES data is created.
- 2. A baseline deep learning model for the automatic delineation of the ice bottom and the ice surface is proposed.
- 3. A thorough evaluation of individual models and an omni-model is conducted on the dataset.

The work is structured as follows: Section 2 provides an overview of datasets and algorithms previously used for automatic ice boundary extraction. Subsequently, Sect. 3 gives insight into the recording and processing of the dataset as well as relevant geographical and glaciological factors of the study sites. The baseline models are introduced in Sect. 4. An extensive evaluation of the baseline models and the benchmark dataset is presented in Sect. 5. Lastly, we summarize our research and draw conclusions in Sect. 6.

**2 Related Works**

Over the past decades, RES has been widely used in glaciology. A multitude of publications cover the extraction of ice boundary layers from RES data. In this section, we highlight related RES datasets and layer extraction approaches.

**210 **2.1** Datasets**

200

205

215

220

Numerous RES datasets on glaciers and ice sheets are publicly available. However, a large portion of the associated ice bottom labels are inaccurate, generated automatically, unavailable, lack the necessary transparency in their creation, or do not have associated radargrams (Young et al., 2021; Blankenship et al., 2018; CReSIS; Dong et al., 2022; Corr, 2020; Corr et al., 2020). This makes them unsuitable for training or evaluating deep learning approaches, as they require human-annotated data. Hence, we focus our comparison on datasets that have been used to extract the ice boundary in previously published work and for which both radargrams and human-annotated labels are publicly available. However, most existing datasets do not meet the requirements necessary for benchmarking models. Common issues include that most datasets are either missing the layer labels, radargrams, or are not publicly available (Young et al., 2021; Blankenship et al., 2018; Dong et al., 2022; Corr, 2020). As a result, accurately benchmarking models to extract the ice bottom often becomes unfeasible. From the datasets, where both the labels and the radargrams are publicly available, a large portion contains labels that are generated automatically with human

supervision (Corr et al., 2021; CReSIS). While this technique is sufficient for clearly visible layers like the ice surface, the ice bottom is often too ambiguous for an automatic tracker to capture accurately in its entirety. Thus, the supervising human would have to intervene in such cases. However, the level of human involvement and the methods to validate the picking process are rarely specified, creating a level of uncertainty that undermines the reliability of the benchmark dataset. Lastly, some datasets also do not satisfy the qualitative standards. While the picking process can often be subjective, errors like a negative ice thickness are an indication of a lack of accuracy, making them unsuitable for training or evaluating deep learning approaches (CReSIS, 2012, 2013). Since listing and evaluating every dataset is virtually impossible, we focus our comparison on datasets that have been used to extract the ice boundary in previously published work and for which both radargrams and human-annotated labels are publicly available. These constraints significantly limit the number of related datasets.

-C4 (2.6)

The one RES system that has been used extensively to collect such data is the Multichannel Coherent Radar Depth Sounder versions 1-5 (MCoRDS) (Allen et al., 2012a), which was used, for example, in NASA's Operation IceBridge (OIB) program on a McDonnell Douglas DC-8-72 jetliner (Shi et al., 2010a). The data acquired over Antarctica in 2009 are the most widely used (Crandall et al., 2012; Lee et al., 2014; Rahnemoonfar et al., 2017a, b; Berger et al., 2018; Kamangir et al., 2018). However, also data from different years (Kamangir et al., 2018; Mitchell et al., 2013; Cai et al., 2020; García et al., 2021a, b; Cai et al., 2022, 2019; Ghosh and Bovolo, 2022; García et al., 2023; Donini et al., 2022; Ilisei and Bruzzone, 2014, 2015) and other locations like Greenland (Donini et al., 2022) and the Canadian Arctic Archipelago (Xu et al., 2017, 2018) were analyzed.

Only very few publications included data from RES systems other than MCoRDS. Gifford et al. (2010) extracted the ice boundary from data acquired by a predecessor RES system (Lohoefener, 2006) during 2006 and 2007 in Greenland. Dong et al. (2022) featured data from the Chinese Academy of Sciences' Deep Ice Radar acquired during the 29th Chinese Antarctic Scientific Expedition. Lastly, Liu-Schiaffini et al. (2022a) used algorithm-assisted human-labeled data acquired in the Canadian Arctic and Antarctica by the University of Texas Institute for Geophysics' high-capability radar sounder (HiCARS). A major downside of these datasets is that they do not provide standardized training and evaluation splits, making inter-model comparison challenging. Furthermore, datasets usually only focus on a single area, e. g., Greenland or Antarctica, which makes generalization to other areas or glaciological settings difficult to verify. IceAnatomy addresses this issue by including data from multiple study sites, radar systems, and glaciological settings. It also provides standardized splits for training and evaluation to allow for an accurate and fair comparison between models. In conclusion, to the best of our knowledge, there is no comparable benchmark dataset for ice boundary extraction from radio-echo-sounding data.

**2.2 Algorithms**

225

230

235

240

245

RES has been employed to detect crevasses (Liu et al., 2020; Walker and Ray, 2019; Williams et al., 2012, 2014), the ice boundary (Crandall et al., 2012; Lee et al., 2014; Rahnemoonfar et al., 2017a, b; Berger et al., 2018; Kamangir et al., 2018; Mitchell et al., 2013; Xu et al., 2017, 2018; Cai et al., 2022; Gifford et al., 2010; Dong et al., 2022; Liu-Schiaffini et al., 2022a), to segment subsurface structures (Cai et al., 2020, 2019; García et al., 2021a, b; Ghosh and Bovolo, 2022; García et al., 2023; Donini et al., 2022; Ilisei and Bruzzone, 2014, 2015), and to track internal ice and snow layers (Crandall et al., 2012; Karlsson

et al., 2013; Ibikunle et al., 2020; Rahnemoonfar et al., 2021; Varshney et al., 2020, 2021; Yari et al., 2019, 2020; Dong et al., 2022). Mogadam and Eisen (Mogadam and Eisen, 2025) provide an overview of the methods used in this domain.

To obtain the ice boundary, one can either directly delineate the ice surface and bottom or first segment different regions such as ice, bedrock, and air and then extract the two layers during post-processing. Most existing studies (Crandall et al., 2012; Lee et al., 2014; Rahnemoonfar et al., 2017a, b; Berger et al., 2018; Kamangir et al., 2018; Mitchell et al., 2013; Xu et al., 2017, 2018; Cai et al., 2022; Gifford et al., 2010; Dong et al., 2022; Liu-Schiaffini et al., 2022a) prefer direct extraction. Fewer studies (Cai et al., 2020, 2019; García et al., 2021a, b; Ghosh and Bovolo, 2022; García et al., 2023; Donini et al., 2022; Ilisei and Bruzzone, 2014, 2015) use the segmentation approach. The segmentation approach assigns a semantic class to each pixel in the radargram, from which the ice boundaries can be derived directly or after post-processing.

In terms of methodology, early studies mainly used classical image processing and machine learning techniques such as

Hidden Markov Models (Crandall et al., 2012; Berger et al., 2018), Markov-Chain Monte Carlo (Lee et al., 2014), contour
detection (Rahnemoonfar et al., 2017a), the level set approach (Rahnemoonfar et al., 2017b; Mitchell et al., 2013), Markov
Random Fields (Xu et al., 2017), edge-based and active contour methods (Gifford et al., 2010), Kullback-Leibler maps (Ilisei
and Bruzzone, 2014), and Support Vector Machines (Ilisei and Bruzzone, 2015). After 2017, studies turned to Convolutional
Neural Networks (CNNs) (Kamangir et al., 2018; Cai et al., 2020, 2019; García et al., 2021a; Cai et al., 2022; Donini et al.,
2022; Dong et al., 2022; Liu-Schiaffini et al., 2022a; García et al., 2021b, 2023; Jebeli et al., 2023a, b; Matsuoka et al., 2021;
Moqadam et al., 2025), combinations of CNNs and Recurrent Neural Networks (RNNs) (Xu et al., 2018), and combinations
of CNNs and Transformers (Ghosh and Bovolo, 2022).

In comparison, we rely on the U-Net architecture from (Ronneberger et al., 2015) to evaluate our newly created dataset. Furthermore, we integrate Atrous Spatial Pyramid Pooling from (Chen et al., 2018) and the ResBlock design from (Esser et al., 2020) to improve the performance.

**3 Dataset**

275

280

260

In this section, we introduce the benchmark dataset "IceAnatomy" which covers several different geolocations and was acquired by multiple radar systems. We divide the dataset into three subsets based on the sources of the data: the AWI (Alfred Wegener Institute, Helmholtz Centre for Polar and Marine Research), CReSIS (The Center for Remote Sensing and Integrated Systems), and FAU (Friedrich-Alexander-Universität Erlangen-Nürnberg, Institute of Geography) subsets. A summary of the most important information about the dataset is given in Tab. 1.

**3.1 Study Sites**

**3.1.1 Southern Patagonian Icefield**

The Southern Patagonian Icefield (SPI) is the largest temperate ice body in the Southern Hemisphere. It is characterized by one of the highest mass loss rates in the world (Zemp et al., 2019; Marzeion et al., 2018; Hugonnet et al., 2021) and by its

**Table 1.** A summary of details about the IceAnatomy benchmark dataset (Lippl et al., 2019; Shi et al., 2010b; Rückamp and Blindow, 2012; CReSIS, 2024a; Allen et al., 2012b; Steinhage, 2001, 2015). VR stands for the vertical resolution of the two-way travel.

|        | Study Sites         | VR                                    | Width-Reso.                 | Length             | Year    | Main Thermal | Labeled  |
|--------|---------------------|---------------------------------------|-----------------------------|--------------------|---------|--------------|----------|
|        |                     |                                       |                             |                    |         | Regime       | Bottom % |
|        | James Ross Island   | $2.5\mathrm{ns}\mathrm{pixel}^{-1}$   | $2\mathrm{mpixel}^{-1}$     | 275 km             | 2017/18 | Polythermal  | 82.5%    |
| FAU    | Perito Moreno       | $2.5\mathrm{ns}\mathrm{pixel}^{-1}$   | $2\mathrm{mpixel}^{-1}$     | $145\mathrm{km}$   | 2022    | Temperate    | 83.1%    |
|        | Viedma              | $2.5\mathrm{ns}\mathrm{pixel}^{-1}$   | $2\mathrm{mpixel}^{-1}$     | $140\mathrm{km}$   | 2022    | Temperate    | 46.2%    |
| CReSIS | Antarctic Peninsula | $105\mathrm{ns}\mathrm{pixel}^{-1}$   | $12\mathrm{mpixel^{-1}}$    | 20400 km           | 2009    | Polythermal  | 63.9%    |
|        | West Antarctica     | $105\mathrm{ns}\mathrm{pixel}^{-1}$   | $12-30\mathrm{mpixel^{-1}}$ | $24400\mathrm{km}$ | 2009    | Polythermal  | 78.9%    |
| AWI    | Antarctic Peninsula | $12\mathrm{ns}\mathrm{pixel}^{-1}$    | $62\mathrm{mpixel^{-1}}$    | 1490 km            | 2013    | Polythermal  | 31.7%    |
|        | East Antarctica     | $13.33\mathrm{ns}\mathrm{pixel}^{-1}$ | $66-79\mathrm{mpixel^{-1}}$ | $1015\mathrm{km}$  | 1997/99 | Cold-based   | 73.7%    |

large outlet glaciers that drain into lakes or the ocean (Aniya, 1999). Two of the largest eastward-flowing outlet glaciers in the region are the Perito Moreno and Viedma glaciers. The only way to obtain information over large areas about their bedrock topography is by helicopter-borne RES measurements. This is particularly applicable to the lower parts of the glaciers, which are surrounded by steep mountain flanks and have heavily crevassed surfaces. As the glaciers are temperate, i. e., most of the ice is close to or at the pressure melting point, they contain a relatively high proportion of water (Aristarain and Delmas, 1993; Millan et al., 2019; Strelin et al., 2014; Schaefer et al., 2015). This characteristic, combined with the steep and deep [A3 (2.9)] 
[revised manuscript text omitted]

The initial layer labels were fully-manually annotated by a single interpreter to ensure consistency throughout the dataset. Surface reflections were generally straightforward to identify; however, in heavily crevassed areas, we increased the resolution to delineate the air-ice interface as accurately as possible across these features. Bedrock picks were conducted using the same approach. In regions with ambiguous reflections, ReflexW (Sandmeier, 2024) software enabled zooming into specific subsets of the radargrams, thereby enhancing the clarity of features of interest. Additionally, several intersecting profile lines provided cross-points for internal validation. These intersections were annotated independently by the same interpreter and subsequently compared. All cross-profile values fell within the expected margin of error, even in areas with steep slopes or greater depths (i.e., deviations < 10 %). At Glacier Perito Moreno, two control points from previous studies were available for comparison (Sugiyama et al., 2011; Stuefer et al., 2007). The first, along the 'Buscaini' profile, corresponds to a seismic survey conducted in 1996, which reported a maximum ice thickness of 720 m. The second, located nearer to the glacier terminus, corresponds to a borehole drilled in 2010, revealing an ice thickness of 515 ± 5 m. Both control points were in close agreement with our ice thickness estimates.

**3.2.2 CReSIS Data**

Three were sea-ice surveys and thus are not included in the CReSIS dataset. The remaining 18 missions can be split into two groups: six missions focusing on the Antarctic Peninsula (PEN1, PEN2, PEN3, PEN4, PEN5, and LVISPEN) and 12 missions exploring West Antarctica (PIG1, PIG2, PIG3, PIG4, LVISPIG, LVIS86, GETZ1, ABBOTT1, TSK1, TSK2, TSK3, and TSK4) (Allen et al., 2012b). All 18 missions employed the Multichannel Coherent Radar Depth Sounder (MCoRDS) flown on a McDonnell Douglas DC-8-72. It has a center frequency of 195 MHz and an eight-channel-chirp signal to accurately assess the ice (Rodriguez-Morales et al., 2014; Shi et al., 2010b).

To process the recorded data, the standard CReSIS L1B CSARP-mvdr (minimum variance distortionless response) processing steps were applied. These include pulse compression via a Tukey and Hanning Window, beam-forming, motion compensation, synthetic aperture radar processing in combination with f-k migration, channel combination, and waveform combination (CReSIS, 2024b). After the processing, the radargrams had a depth-resolution vertical resolution of the two-way travel time (VR) of 105 ns pixel-1 and a width resolution of 12 - 30 m pixel-1 depending on the mission.

C5 (2.15)

We obtained the fully processed CReSIS subset by downloading the CSARP-mvdr processed L1B product from the CReSIS website and taking the square root of the amplitudes. Likewise, CReSIS also provides downloads for the annotated ice bottom and surface layers on their website (CReSIS, 2024a). According to Lee et al. (2014) and Crandall et al. (2012), the rock-bed surface is humanly annotated but noisy. Although the noise might pose a problem for certain approaches, we chose not to alter the labels. The reason for this is that the dataset has been used previously in other publications, and in order to remain comparable, we use the same labels. However, to provide additional context regarding the quality, CReSIS provides a quality label for every pick. The label indicates the annotator's confidence, ranging from one (high) to three (low). We include these labels in the benchmark dataset for future research. The general picking procedure for CReSIS data is outlined in (CReSIS, 2024b).

**3.2.3 AWI Data**

380

385

390

395

The AWI subset was recorded during campaigns in Dronning Maud Land in 1997 and 1999 (Steinhage et al., 2023b, a) and in the Antarctic Peninsula in November 2013 (Steinhage, 2015). All three campaigns employed a version of the EMR radar system with a center frequency of 150 MHz and the toggle mode enabled. The toggle mode alternates the radar's pulse length between 60 ns and 600 ns periodically. Thus, the system can achieve a decent depth-resolutionVR while capturing deep internal layers of the ice. The processing of the recorded data was similar for all three campaigns. The data was differentiated, rescaled, high-pass filtered, and bandpass filtered. To reduce the amount of noise in a radargram, multiple traces were combined into a single trace. In detail, ten traces were combined for the 1997 and 1999 flights, and seven traces were combined for the 2013 flight (Steinhage, 2001; Nixdorf et al., 1999; Steinhage et al., 2001). Automatic gain control was used to normalize the amplitude values. After the processing, the radargrams had a depth-resolutionVR of 12 – 13.33 ns pixel-1 and a width resolution of 66 – 79 m pixel-1 depending on the campaign.

The ice surface and ice bottom were annotated by one person. To ensure consistency, plausibility checks were performed at crossing points with other profiles from the same or related campaigns. No systematic biases were observed. In the picks, gaps of eleven pixels or less were filled using bicubic interpolation. Finally, for the radargrams from 1997 and 1999, all data below 3600 pixels, which is about 4 km, was discarded because only noise was visible at these depths. The gathered data was processed with FOCUS, DISCO, LANDMARK, and Python.

**4 Baseline Method**

[revised manuscript text omitted]

error (MSE) (Crandall et al., 2012; Mitchell et al., 2013; Dong et al., 2022), the root mean square error (RMSE) (Liu-Schiaffini et al., 2022a), and the largest under- and over-estimation (Gifford et al., 2010). We can also define confusion matrix-based metrics on the layer extraction task. In that case, we define each height pixel of the radargram as a separate class, and the closest pixel in each column to the corresponding layer as the correct class. However, a limitation of confusion matrix-based metrics, such as precision, is that they do not account for distance weighting. For example, if a prediction is always one pixel next to the annotated layer, also known as ground truth (GT), the confusion matrix-based metrics will have the worst possible value, even though it is a near-perfect prediction. Therefore, some studies (Xu et al., 2017; Gifford et al., 2010; Liu-Schiaffini et al., 2022a) have relaxed these confusion matrix-based metrics by considering predictions a few pixels from the ground truth as still correct.

480

**Figure 4.** A visual representation of the four metrics used in this work. The left side of the figure depicts the MAE and MME respectively as the difference between prediction and ground truth (GT). Meanwhile, the right side of the figure features the AP-1% and AP-5% respectively as an interval around the ground truth. Note that the ground truth and the predictions are technically float numbers. However, we thickened the ground truth by 20 pixels to improve visibility.

As metrics for our benchmark framework, we have chosen the MAE, two relaxed Average Precision (AP) metrics, and introduce the Mean Meter Error (MME). The MAE is calculated as the column-wise difference in pixels between the ground truth depth of a layer and the predicted depth. Resizing the radargram will change the value of this metric. Therefore, we also introduce the MME, which approximates the real-world error. We calculate the MME by multiplying the MAE with the product of the wave velocity in the medium and the depth resolution VR of the radargram. The wave velocity describes

the speed of the electromagnetic wave of the radar through a medium. We assume it to be constant with the speed of light  $(c_{air} = 0.299792458 \,\mathrm{m\,ns^{-1}})$  in air and with  $c_{ice} = 0.168 \,\mathrm{m\,ns^{-1}}$  in ice (Johari and Charette, 1975). The depth resolution VR is  $[c_{air} = 0.299792458 \,\mathrm{m\,ns^{-1}}]$  in air and with  $c_{ice} = 0.168 \,\mathrm{m\,ns^{-1}}$  in ice (Johari and Charette, 1975). The depth resolution VR is indirectly proportional to the y-dimension of the radargram, the MME stays consistent across different heights. Table 1  $[c_{air} = 0.299792458 \,\mathrm{m\,ns^{-1}}]$  records the different depth resolutions VRs for radargrams in the IceAnatomy dataset in their original height and equation 3  $[c_{air} = 0.299792458 \,\mathrm{m\,ns^{-1}}]$  and 4 summarize the formula for the MME. Note that the MME is still highly dependent on the original depth resolution VR of  $[c_{air} = 0.299792458 \,\mathrm{m\,ns^{-1}}]$  and 4 summarize the formula for the MME. Note that the MME is still highly dependent on the original depth resolution VR of  $[c_{air} = 0.299792458 \,\mathrm{m\,ns^{-1}}]$  the radargram. The MME will be naturally higher for a radargram where every pixel constitutes a 40 m change in height rather than a 4 m change, as even small mistakes lead to a drastic increase. Thus, we also record the MAE as it is more consistent over radargrams of the same image height but with different study sites and radar systems. For more details on the VR, compare Appendix E.

[revised manuscript text omitted]

combinations. Still, dataset and model-specific discrepancies exist.

**5.3.1** Ice Surface Predictions**

The predictions for the ice surfaces are nearly perfect for all subsets and all models. The three subset models even achieve 100 % accuracy for the AP-5%. Hence, the remaining discrepancies are likely significantly influenced by measurement inaccuracies, noise, and general model variance. Therefore, we will only consider the task of ice bottom delineation to assess model performance.

**5.3.2** Ice Bottom Predictions**

For the ice bottom predictions, the differences in the MME between the three subsets are more pronounced than for the MAE,

which can be attributed to the different depth resolutions VRs. The MAE difference between the FAU and CReSIS subsets is small, while the MAE on the AWI subset is substantially lower than both. The AP-1% is lower for the FAU subset than for the AWI and CReSIS subsets. Interestingly, this difference between subsets is relativized for AP-5%. This means that most incorrect predictions for FAU are in the 1% to 5% error range. The same is true for the AWI subset. For the CReSIS data, this effect is not as strong. Here, the AP only increases from 87.9% for the 1% error rate to 94.1% for the 5% error rate.

**550 **5.3.3** Omni Model**

555

The omni model shows persistently higher MME and MAE values and lower AP-1% and AP-5% values for the FAU and AWI subsets than the dataset-specific models. In detail, it only achieves an MME of 19.5 m and 39.8 m and an AP-1% of 68.3% and 75.7%, respectively. We attribute the lower performance of the omni model to the substantial domain shift between the three subsets and the fact that the FAU and AWI subsets are significantly smaller than the CReSIS subset. For the CReSIS subset, the omni model outperforms the dataset-specific model. In particular, it achieves an MME of 66.5 m and an AP-1% of 88.6%. These results suggest that there can be a benefit from more training data even with the domain shift. However, the domain shift makes the generalization to under-represented or new domains difficult.

**5.3.4** Influence of Study Sites**

Table 3 divides the results of the subset-specific models by study site and thermal regime.

For the FAU subset, the Perito Moreno and Viedma predictions are quantitatively worse than the ones from JRI. A key difference between Perito Moreno, Viedma, and JRI is the thermal regime. The first two are temperate glaciers, while JRI contains polythermal ice. Besides the higher water content in Perito Moreno and Viedma, both are also substantially deeper than JRI in most areas. They even have areas with ice thicker than the 700 m maximum penetration depth of the employed radar system. Viedma and JRI also feature several-meter-thick moraine material on the glacier surface. These rock and debris deposits are not penetrable by the wavelets and thus create radar shadows below them or substantially decrease the amount of reflected energy.

**Table 3.** Overview of the influence of geographical and glaciological factors on the performance in detecting the ice bottom. We differentiate between the subset, the study site, and the general thermal regime. For the performance analysis, we compare the MME, MAE, AP-1%, and AP-5% as defined in Section 5.1. To contextualize the MME, we annotate the relative error to the mean measured ice thickness of the specified test set study site behind the MME. Note that for the AWI subset-specific model, we utilized the weights of the omni model as a starting point to stabilize training. We conducted the evaluation on the test set and averaged the results over five runs to minimize statistical errors. The analyzed models were the subset-specific models.

|        | Study Site          | Main Thermal Regime | $\mathbf{MME}\downarrow$  | MAE ↓ | AP-1% ↑ | AP-5% ↑ |
|--------|---------------------|---------------------|---------------------------|-------|----------------|----------------|
|        | Perito Moreno       | Temperate           | 22.1 m [8.0 %]            | 26.3  | 54.9%          | 91.1%          |
| FAU    | Viedma              | Temperate           | $10.0\mathrm{m}\ [5.0\%]$ | 12.0  | 68.5%          | 96.8%          |
|        | James Ross Island   | Polythermal         | $3.9\mathrm{m}\;[2.7\%]$  | 9.2   | 84.9%          | 96.9%          |
| CReSIS | Antarctic Peninsula | Polythermal         | 31.6 m [4.5 %]            | 5.8   | 91.5%          | 97.6%          |
|        | West Antarctica     | Polythermal         | $148.7\mathrm{m}[18.0\%]$ | 29.4  | 82.5%          | 88.8%          |
| AWI    | Antarctic Peninsula | Polythermal         | 32.7 m [9.8 %]            | 8.1   | 87.3%          | 96.4%          |
|        | East Antarctica     | Cold-based          | $27.3\mathrm{m}\;[1.0\%]$ | 6.9   | 81.1%          | 98.3%          |

If we look at the associated radargrams, we can mostly see a relatively stable and clear prediction for JRI. On the other hand, Viedma and Perito Moreno have much stronger differences to the ground truth. Especially in deep and noisy regions, the models struggle. Figure 5 shows example traces for the three study sites of the FAU subset.

Between the Antarctic Peninsula and West Antarctica study sites of the CReSIS subset, there are strong differences in the quantitative analysis. The MME and MAE values exhibit a difference of approximately a factor of five, while the AP-1% and AP-5% are approximately 9% apart. In the qualitative analysis, we can see that the predictions in both regions actually follow the ground truth closely. However, sometimes the predicted ice bottom layer makes a jump and the actual ice surface is predicted to be the ice bottom. We call this "ice boundary collapse". Examples of this phenomenon can be seen in Fig. 6. For the AWI subset, the results for East Antarctica are generally more favorable than those for the Antarctic Peninsula. This result is consistent with the observation on the FAU subset that the algorithm performs better for colder ice than for warmer ice. The only exception is the AP-1%, where the Antarctic Peninsula slightly outperformed East Antarctica. This result suggests that a large majority of the wrong predictions in East Antarctica are between the 1% and 5% interval and that our algorithm struggles to pinpoint the exact location of the ice bottom. We can confirm this behavior in the qualitative analysis, where the prediction is sometimes slightly above or below the ground truth line but follows it closely overall. Similarly, the predictions for the Antarctic Peninsula also appear to be very accurate but contain more occasional outliers. Figure 7 depicts both the predictions for East Antarctica and the Antarctic Peninsula.

**5.3.5 Loss Function**

570

575

580

585

To assess the performance of our combined loss function, we conducted a small ablation study. Specifically, we evaluated two additional experiments in which we replaced the combined loss with each of its individual components: In the first setup, we

**Figure 5.** Visualization of the subset-specific model's performance on the FAU subset. Figure (a) shows trace 3000-5500 of the third flight over Perito Moreno, Fig. (b) depicts traces 5000-7500 of the second flight over Viedma, and Fig. (c) presents traces 5000-7500 from the first Section of the 2017 flights over James Ross Island.

**Figure 6.** Visualization of the subset-specific model's performance on the CReSIS subset. Figure (a) presents traces 2000-4500 from mission PEN4 in the Antarctic Peninsula (PEN4\_01\_001). Figure (b) presents traces 2000-4500 from mission TSK2 in West Antarctica (TSK2\_07\_003).

**Figure 7.** Visualization of the subset-specific model's performance on the AWI subset. Figure (a) depicts traces 21000-23500 from the 2014 flight in the Antarctic Peninsula. Figure (b) presents traces 7837-9787 from the 1999 flight in East Antarctica.

**Table 4.** Summary of our ablation study regarding the proposed modifications to the loss function. For every variation of the loss function, we trained a subset-specific model and compared the performance based on the MME and AP-1% of the ice bottom layer. We conducted the evaluation on the test set and averaged the results over five runs to minimize statistical errors. To contextualize the MME, we annotate the relative error to the mean measured ice thickness of the specified test set study site behind the MME. Note that for the AWI subset-specific model, we utilized the weights of the omni model as a starting point to stabilize training, which was also trained with the specified loss function.

|                                       | FAU                      |                | CReSIS                      | S              | AWI                       |                |  |
|---------------------------------------|--------------------------|----------------|-----------------------------|----------------|---------------------------|----------------|--|
|                                       | $\mathbf{MME}\downarrow$ | AP-1% ↑ | $\mathbf{MME}\downarrow$    | AP-1% ↑ | $\mathbf{MME}\downarrow$  | AP-1% ↑ |  |
| $L_{\text{CE}}$                       | 13.9 m [7.4 %]           | 74.3%          | 88.0 m [11.7%]              | 88.6%          | 29.7 m [1.6 %]            | 82.5 %         |  |
| $L_{ m Dist}$                         | $9.9\mathrm{m}\;[5.1\%]$ | 72.3%          | $119.5\mathrm{m}\;[15.9\%]$ | 85.5%          | $33.2\mathrm{m}[1.7\%]$   | 81.8%          |  |
| $L_{\mathrm{CE}} + L_{\mathrm{Dist}}$ | $9.1\mathrm{m}\;[4.9\%]$ | 74.3%          | $78.2\mathrm{m}[10.4\%]$    | 87.9%          | $29.3\mathrm{m}\;[1.5\%]$ | 83.5%          |  |

trained the model with the cross-entropy loss, and in the second setup, we trained it only with the distance loss. We compare the results of these two configurations with the combined loss in Table 4.

[revised manuscript text omitted]

Previous work has already demonstrated the effectiveness of automatic approaches for ice boundary extraction but lacked a common method for accurately comparing models. With this benchmark framework, we hope to address this issue by unifying and standardizing both training and evaluation schemes. We hope that this benchmark dataset will encourage more scientists to engage in this challenging and important research area. Deep learning models that extract the ice boundary can greatly speed up the processing of RES data. As a result, the ice thickness and, consequently, the subglacial topography can be determined more quickly after a field survey.

*Code and data availability.* The dataset is available at https://zenodo.org/records/14036897 (Dreier et al., 2024) and the implementation at https://doi.org/10.5281/zenodo.14038570 (Dreier, 2024).

**Appendix A: Additional Hyperparameters**

This section gives an overview of the hyperparameters in our employed U-Net from Chapter 4.2. The input dimension of our U-net is (1024,512,1) (H,W,1), which then gets scaled according to the depth level of the encoder or decoder. Inside the network, we down- and upsample our feature map five times each while scaling the feature dimension according to the depth-level-dependant value of [8,16,32,64,64,128]. To reduce the risk of overfitting, we also utilize dropout layers inside the ResBlocks with a probability of 10%. For the loss function, we employed our proposed combined loss function. Since the numerical value of the distance loss is significantly higher than that of the classification loss, we had to weigh the individual components. In detail, we chose the weights  $w_{\text{s\_class}} = 0.5$ ,  $w_{\text{b\_class}} = 1.0$ ,  $w_{\text{s\_dist}} = 0.05$ , and  $w_{\text{b\_dist}} = 0.1$  as they performed the best in preliminary experiments.

**Appendix B: ResBlock Design**

670

675

To provide a better understanding of the network architecture, this section examines one of its core components: the ResBlock from (Esser et al., 2020). Its structure, shown in Fig. B1, comprises several components. First, it starts with a group normalization layer (Wu and He, 2018) that normalizes the data in groups of channels to increase stability during training. Next, a swish activation (Ramachandran et al., 2017) function adds nonlinearity to the ResBlock so the network can learn more complex patterns. The activation is followed by a two-dimensional convolution layer that processes and combines the visual features by applying convolutional operations. This is followed by another group normalization and swish activation function before a regular dropout layer (Hinton et al., 2012) is applied. The dropout layer randomly withholds information during training to improve generalization and prevent the model from overfitting – a process in which the model develops a strong bias towards the training data. After the dropout layer, another two-dimensional convolutional layer is applied. Finally, a residual connection (He et al., 2016), a shortcut from the start of the ResBlock to the end through a convolution layer, is added to the output of this sequence of layers to improve the gradient flow in the network.

**680 Appendix C: Loss Function Details**

The formulas of the classification and distance-based loss are as follows:

$$L_{\text{CE}} = -\sum_{c \in C} x_c (1 - \epsilon_c) \log(p(x_c)) + \frac{\epsilon_c (1 - x_c) \log(p(x_c))}{|C|}$$
(C1)

$$L_{\text{class}} = \frac{w_{\text{s\_class}}}{|\hat{Y}_{\text{s}}|} \sum_{\hat{y}_{\text{s}} \in \hat{Y}_{\text{s}}} L_{\text{CE}}(\hat{y}_{\text{s}}) + \frac{w_{\text{b\_class}}}{|\hat{Y}_{\text{b}}|} \sum_{\hat{y}_{\text{b}} \in \hat{Y}_{\text{b}}} L_{\text{CE}}(\hat{y}_{\text{b}})$$
(C2)

$$L_{\text{dist}} = \frac{w_{\text{s\_dist}}}{|\hat{Y}_{\text{s}}|} \sum_{\hat{y}_{\text{s}} \in \hat{Y}_{\text{s}}} \langle d(y_{\text{s}}), \sigma(\hat{y}_{\text{s}}) \rangle + \frac{w_{\text{b\_dist}}}{|\hat{Y}_{\text{b}}|} \sum_{\hat{y}_{\text{b}} \in \hat{Y}_{\text{b}}} \langle d(y_{\text{b}}), \sigma(\hat{y}_{\text{b}}) \rangle \tag{C3}$$

685  $w_{\text{s\_class}}, w_{\text{b\_class}}, w_{\text{s\_dist}}$ , and  $w_{\text{b\_dist}}$  are the respective weights for a weighted combination of the single loss parts,  $\epsilon_c$  is the smoothing factor, C specifies the column,  $\langle \rangle$  is the dot product,  $\sigma$  is the softmax function that converts the model's outputs into

\* skipped if input and output have the same shape

**Figure B1.** Structure of the residual block employed in our deep learning model. The arrangement is based on the design of Esser et al. (2020)

probabilities, |.| is the cardinality of a set, and d is the function that creates a vector filled with the column-wise distance map given the respective column of the label.

**Appendix D: Additional Experiments**

Since the three subsets of IceAnatomy differ in size, we also investigate whether a uniform sampling strategy, where samples are drawn equally from each subset, could help the Omni-Model achieve the performance of the domain-specific models on the AWI and FAU subsets. From our results in Table D1, we can see that a uniform sampling strategy does lead to improvement for the AWI and FAU subsets. In the case of the AWI subset, the omni model even outperforms the domain-specific model. However, in the case of the FAU subset, we are still below the domain-specific model. We reason that the domain of the AWI and CReSIS subsets are significantly closer than the FAU subset as these two subsets contain differentiated radargrams. We, therefore, believe that domain shift remains an important area for future research. In addition to the uniform sampling, we also investigated how different hyperparameter setups regarding learning rate and regularization would affect the benchmark model. From the results in Table D2 and D3, we can see that different hyperparameter setups favor different subsets of IceAnatomy. However, there seems to be no universal optimal setup.

**700 Appendix E: Translations of the vertical resolutions**

While the vertical resolution of the two-way travel time approximates the granularity of the specific radargrams, it is challenging to interpret for a real-world scenario. To simplify interpretation, we provide a simple conversion in Table E1 given a fixed

**Table D1.** Overview of the performance of our Omni Model with uniform sampling. We distinguish the layer prediction into two classes: the ice surface (S) and the ice bottom (B). We compare the model's performance on the MME, MAE, AP-1%, and AP-5% as defined in Section 5.1. To contextualize the MME, we annotate the relative error to the mean measured ice thickness of the specified test set study site behind the MME. We conducted the evaluation on the test set and averaged the results over five runs to minimize statistical errors.

|        |       | Omni Model                 |       |                 |                |  |  |  |
|--------|-------|----------------------------|-------|-----------------|----------------|--|--|--|
|        | Layer | $\mathbf{MME}\downarrow$   | MAE ↓ | AP-1 % ↑ | AP-5% ↑ |  |  |  |
| FAU    | S     | 2.0 m [1.1 %]              | 1.9   | 99.3%           | 100.0%         |  |  |  |
|        | В     | $14.0\mathrm{m}~[7.6\%]$   | 19.0  | 74.1%           | 94.1%          |  |  |  |
| CReSIS | S     | 23.1 m [3.1 %]             | 2.5   | 97.2%           | 100.0%         |  |  |  |
|        | В     | $75.0\mathrm{m}\;[10.0\%]$ | 14.6  | 87.7%           | 93.9%          |  |  |  |
| AWI    | S     | 3.8 m [0.2 %]              | 0.5   | 99.7, %         | 100.0%         |  |  |  |
|        | В     | $23.9\mathrm{m}[1.3\%]$    | 6.0   | 86.1%           | 98.3%          |  |  |  |

Table D2. Overview of the performance of our Omni Model with different learning rates and uniform sampling. We distinguish the layer prediction into two classes: the ice surface (S) and the ice bottom (B). We compare the model's performance on the MME and AP-1% as defined in Section 5.1. To contextualize the MME, we annotate the relative error to the mean measured ice thickness of the specified test set study site behind the MME. We conducted the evaluation on the test set and averaged the results over three runs to minimize statistical errors. Note that for lr = 0.005 we averaged over five runs, as we had those values from previous experiments.

|        |       | lr = 0.0001                 |                | lr = 0.0005                |                | lr = 0.001                |                |
|--------|-------|-----------------------------|----------------|----------------------------|----------------|---------------------------|----------------|
|        | Layer | $\mathbf{MME}\downarrow$    | AP-1% ↑ | MME ↓                      | AP-1% ↑ | MME ↓                     | AP-1% ↑ |
| FAU    | S     | 2.1 m [1.1 %]               | 99.0%          | 2.0 m [1.1 %]              | 99.3 %         | 2.1 m [1.1 %]             | 99.1%          |
|        | В     | $14.1\mathrm{m}\;[7.6\%]$   | 73.9%          | $14.0\mathrm{m}\;[7.6\%]$  | 74.1%          | $14.3\mathrm{m}\;[7.7\%]$ | 74.1%          |
| CReSIS | S     | 26.2 m [3.5 %]              | 96.7%          | 23.1 m [3.1 %]             | 97.2%          | 21.9 m [2.9 %]            | 97.6%          |
|        | В     | $105.4\mathrm{m}\;[14.0\%]$ | 87.2%          | $75.0\mathrm{m}\;[10.0\%]$ | 87.7%          | $94.9\mathrm{m}[12.6\%]$  | 87.9%          |
| AWI    | S     | 4.4 m [0.2 %]               | 99.5%          | 3.8 m [0.2 %]              | 99.7, %        | 3.7 m [0.2 %]             | 99.6%          |
|        | В     | $26.9\mathrm{m}\;[1.4\%]$   | 86.2%          | $23.9\mathrm{m}\;[1.3\%]$  | 86.1%          | $21.5\mathrm{m}\;[1.1\%]$ | 87.8%          |

**Table D3.** Overview of the performance of our Omni Model with different learning rates and uniform sampling. We distinguish the layer prediction into two classes: the ice surface (S) and the ice bottom (B). We compare the model's performance on the MME and AP-1% as defined in Section 5.1. To contextualize the MME, we annotate the relative error to the mean measured ice thickness of the specified test set study site behind the MME. We conducted the evaluation on the test set and averaged the results over three runs to minimize statistical errors. Note that for dropout = 0.1 we averaged over five runs, as we had those values from previous experiments.

|        |       | dropout = 0.0              |                | dropout = 0.1              |                | dropout =                 | dropout = 0.2  |  |
|--------|-------|----------------------------|----------------|----------------------------|----------------|---------------------------|----------------|--|
|        | Layer | $\mathbf{MME}\downarrow$   | AP-1% ↑ | MME ↓                      | AP-1% ↑ | $\mathbf{MME}\downarrow$  | AP-1% ↑ |  |
| FAU    | S     | 2.0 m [1.1 %]              | 99.2%          | 2.0 m [1.1 %]              | 99.3%          | 2.0 m [1.1 %]             | 99.3%          |  |
|        | В     | $10.2\mathrm{m}\;[5.5\%]$  | 74.2%          | $14.0\mathrm{m}~[7.6\%]$   | 74.1%          | $14.3\mathrm{m}\;[7.7\%]$ | 73.5%          |  |
| CReSIS | S     | $22.5\mathrm{m}[3.0\%]$    | 97.0%          | 23.1 m [3.1 %]             | 97.2%          | 21.5 m [2.9 %]            | 97.6%          |  |
|        | В     | $79.0\mathrm{m}\;[10.5\%]$ | 87.8%          | $75.0\mathrm{m}\;[10.0\%]$ | 87.7%          | $69.2\mathrm{m}\;[9.2\%]$ | 88.5%          |  |
| AWI    | S     | 7.3 m [0.4 %]              | 99.0%          | 3.8 m [0.2 %]              | 99.7%          | 3.8 m [0.2 %]             | 99.5%          |  |
|        | В     | $26.0\mathrm{m}\;[1.4\%]$  | 85.7%          | $23.9\mathrm{m}~[1.3\%]$   | 86.1%          | $24.5\mathrm{m}[1.3\%]$   | 86.6%          |  |

radargram height of 1024. Although the vertical resolution was fixed for each data source, the ratio of pixels to meters varies depending on the flight, the original height of the radargram, the chosen radargram height, and between the air and ice layers. Interestingly, this table also offers an estimate of the lower error bound introduced by discretizing the continuous height values into pixels. Since the introduced model does not interpolate between pixels, decimal pixel heights cannot be represented accurately. The problem gets further amplified by downscaling the images to a lower resolution, as the pixel-to-meter ratio rises proportionally.

**Table E1.** A conversion table that translates pixels to meters given a fixed height of 1024 pixels.

[revised manuscript text omitted]